# FACTS: TABLE SUMMARIZATION VIA OFFLINE TEMPLATE GENERATION WITH AGENTIC WORKFLOWS

## ABSTRACT

Query-focused table summarization requires generating natural language summaries of tabular data conditioned on a user query, enabling users to access insights beyond fact retrieval. Existing approaches face key limitations: table-to-text models require costly fine-tuning and struggle with complex reasoning, prompt-based LLM methods suffer from token-limit and efficiency issues while exposing sensitive data, and prior agentic pipelines often rely on decomposition, planning, or manual templates that lack robustness and scalability. To mitigate these issues, we introduce an agentic workflow, **FACTS**, *a **F**ast, **A**ccurate, and Privacy-**C**ompliant **T**able **S**ummarization approach via Offline Template Generation*. FACTS produces *offline templates*, consisting of SQL queries and Jinja2 templates, which can be rendered into natural language summaries and are reusable across multiple tables sharing the same schema. It enables fast summarization through reusable offline templates, accurate outputs with executable SQL queries, and privacy compliance by sending only table schemas to LLMs. Evaluations on widely-used benchmarks show that FACTS consistently outperforms baseline methods, establishing it as a practical solution for real-world query-focused table summarization.

## 1 INTRODUCTION

Query-focused table summarization requires generating natural language summaries of tabular data conditioned on a user query, enabling users to access insights that go beyond fact retrieval (Zhao et al., 2023). Unlike generic table summarization (Lebret et al., 2016; Moosavi et al., 2021), which aims to capture all salient table content, query-focused summarization adapts to diverse user intents. Compared with table question answering (Pasupat & Liang, 2015; Nan et al., 2022), which typically returns short factoid answers, query-focused summarization demands richer reasoning and explanatory narratives. This distinction is especially critical in real-world domains such as finance, healthcare, and law, where professionals rely on customized summaries for decision-making. For instance, in a financial institution, analysts may request *gross income summaries*, one for each year over the past ten years, providing a *user query* as in Figure 1 (top left).

We argue that a practical solution must handle large datasets efficiently, support reusability, ensure correctness of outputs, and protect sensitive information. These four properties are essential for query-focused table summarization methods in practice. First, the method must be fast, enabling reusability across tables with the same schema and scalability to very large tables without passing all rows to language models. Second, it must be accurate, grounding summaries in executable operations rather than free-form text generation. Third, it must be privacy-compliant, since regulations such as HIPAA and GDPR often prohibit exposing individual-level records to external LLM services.

Yet existing approaches fall short. Table-to-text models (Liu et al., 2022b; Zhao et al., 2022; Jiang et al., 2022) require costly fine-tuning and still struggle with numerical reasoning and logical fidelity. Prompt-based methods (Zhao et al., 2023; Zhang et al., 2024) directly query powerful LLMs but suffer from token-limit and efficiency issues while exposing sensitive data from the tables. Prevalent agentic frameworks (Cheng et al., 2023; Ye et al., 2023; Zhao et al., 2024; Zhang et al., 2025) mitigate some challenges by grounding outputs in SQL or Python execution, but most rely on decomposition, natural language planning, or manual template design, which lack robustness and scalability. Returning to our previous example, an approach such as DirectSumm would require ten separate LLM generations

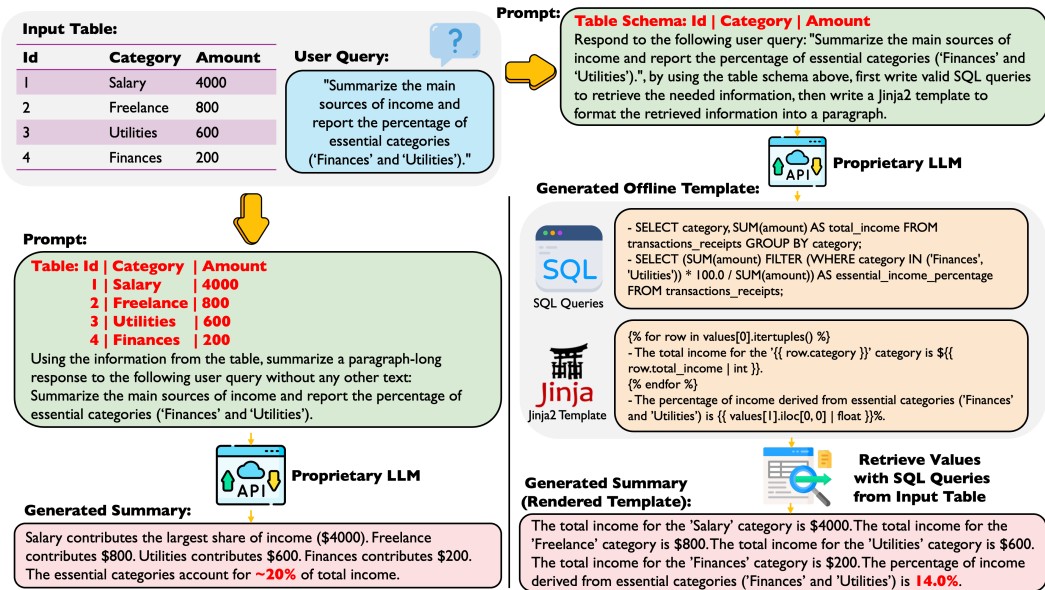

Figure 1: Comparison between DirectSumm (Zhang et al., 2024) (left) and our proposed FACTS framework (right). DirectSumm prompts a large language model (LLM) with the full table and query, which may produce hallucinated values, exposes all table records to external services, and requires regeneration for each new table even under the same schema and query. In contrast, FACTS generates a reusable offline template consisting of schema-aware SQL queries and a Jinja2 template. The SQL queries retrieve precise values through execution, while the Jinja2 template renders natural language summaries, ensuring accuracy, reusability, scalability, and privacy compliance.

for ten yearly tables, with all values revealed to the model, leading to inefficiency and privacy risks, as illustrated in Figure 1 (left).

To address these challenges, we introduce **FACTS**, *a **F**ast, **A**ccurate, and Privacy-**C**ompliant **T**able **S**ummarization approach via Offline Template Generation*. FACTS employs an agentic workflow with three stages. First, it generates schema-aware guided questions and filtering rules to clarify user query intent. Second, it synthesizes SQL queries to extract relevant information from tables. Third, it produces a Jinja2 template to render SQL outputs into natural language. Crucially, FACTS integrates an LLM Council, an ensemble of LLMs iteratively validating and refining outputs at each stage. This feedback loop ensures correctness, consistency, and usability of the generated artifacts. The final product, an offline template composed of SQL queries and a Jinja2 template, can be reused across any tables with the same schema for a given query. Returning to our example, an offline template produced by FACTS can summarize gross income across ten yearly tables, avoiding repeated LLM calls while ensuring accurate and privacy-compliant outputs (Figure 1 (right)). To the best of our knowledge, FACTS introduces the first agentic framework that automates offline template generation for query-focused table summarization.

We evaluate FACTS on three public benchmarks: FeTaQA (Nan et al., 2022), QTSumm (Zhao et al., 2023), and QFMTS (Zhang et al., 2024). Experimental results show that FACTS consistently outperforms representative baselines, demonstrating its practicality for real-world query-focused table summarization.

In summary, our contributions are as follows:

- We propose offline template generation, which produces reusable and schema-specific templates in a privacy-compliant manner, enabling scalability to large tables and efficiency across recurring queries.
- We design FACTS, an agentic workflow that integrates guided question generation, SQL synthesis, and Jinja2 rendering, supported by iterative feedback loops to ensure correctness.
- We demonstrate the practicality of FACTS through comprehensive experiments on FeTaQA, QTSumm, and QFMTS, showing promising improvements over representative baselines.

Table 1: Comparison of paradigms for query-focused table summarization. Only FACTS satisfies all four desired properties. TaPERA and SPaGe produce *partially reusable* plans, denoted as ∼.

| Method | Reusable | Scalable | Accurate | Privacy-Compliant |
|---|---|---|---|---|
| Table-to-Text Models | ✗ | ✗ | ✗ | ✗ |
| Prompt-Based Models | ✗ | ✗ | ✗ | ✗ |
| Binder (Cheng et al., 2023) | ✗ | ✗ | ✓ | ✗ |
| Dater (Ye et al., 2023) | ✗ | ✗ | ✓ | ✗ |
| TaPERA (Zhao et al., 2024) | ∼ | ✗ | ✓ | ✗ |
| SPaGe (Zhang et al., 2025) | ∼ | ✗ | ✓ | ✗ |
| FACTS (ours) | ✓ | ✓ | ✓ | ✓ |

## 2 RELATED WORK

This section reviews prior work related to our study. We first situate query-focused table summarization within the broader landscape of table summarization and question answering. We then survey existing approaches and compare these paradigms against our proposed framework.

**Query-Focused Table Summarization.** Research on table-to-text generation has primarily aimed at transforming structured tables into natural language statements or summaries (Parikh et al., 2020; Chen et al., 2020; Cheng et al., 2022b; Lebret et al., 2016; Moosavi et al., 2021; Suadaa et al., 2021). These works typically target either single-sentence descriptions or domain-specific summaries, with the main goal of improving fluency and factual consistency. However, such outputs are not tailored to a user's specific information needs. In contrast, table question answering (Pasupat & Liang, 2015; Iyyer et al., 2017; Nan et al., 2022) has focused on answering precise fact-based queries, usually returning short values or entities. While table question answering captures query intent, it lacks the ability to provide longer-form reasoning or explanatory summaries. To address this gap, Zhao et al. (2023) introduced the task of query-focused table summarization, where a model generates a narrative-style summary conditioned on both the table and a user query. Compared to generic table summarization, query-focused table summarization explicitly accounts for diverse user intents, and compared to table question answering, it produces extended summaries rather than minimal answers.

**Existing Approaches.** Existing work can be broadly grouped into three categories. **(1)** *Table-to-text models* adapt language models to better capture table structure and reasoning. TAPEX (Liu et al., 2022b) extends BART with large-scale synthetic SQL execution data, improving compositional reasoning. ReasTAP (Zhao et al., 2022) follows a similar idea but uses synthetic QA corpora to enhance logical understanding. OmniTab (Jiang et al., 2022) combines both natural and synthetic QA signals for more robust pretraining. FORTAP (Cheng et al., 2022a) leverages spreadsheet formulas as supervision to strengthen numerical reasoning. PLOG (Liu et al., 2022a) introduces a two-stage strategy: first generating logical forms from tables, then converting them into natural language, to improve logical faithfulness in summaries. **(2)** *Prompt-based models* instead rely directly on large language models (LLMs) with carefully designed prompting. ReFactor (Zhao et al., 2023) extracts query-relevant facts and concatenates them with the query to guide generation. DirectSumm (Zhang et al., 2024) produces summaries in a single step, synthesizing text directly from the table and query. Reason-then-Summ (Zhang et al., 2024) decomposes the task into two stages, first retrieving relevant facts and then composing longer summaries. **(3)** *Agentic frameworks* use external tools such as SQL or Python to ensure accuracy. Binder (Cheng et al., 2023) translates the input query into executable programs, often SQL, to ground results in computation. Dater (Ye et al., 2023) decomposes complex queries into smaller sub-queries, executes them individually, and aggregates their outputs. TaPERA (Zhao et al., 2024) builds natural language plans that are converted into Python programs for execution before aggregation. SPaGe (Zhang et al., 2025) moves beyond free-form plans by introducing structured representations and graph-based execution, improving reliability in multi-table scenarios. Table 1 contrasts our proposed FACTS with representative methods using four criteria. *Reusable*: artifacts applicable to new tables with the same schema; *Scalable*: ability to handle very large tables without feeding all rows; *Accurate*: correctness via executable programs; *Privacy-Compliant*: avoiding exposure of raw table content to LLMs. Most prior methods fall short on one or more dimensions: table-to-text and prompt-based models lack all four; agentic frameworks

Example 1: An offline template generated by FACTS on the QFMTS dataset (Zhang et al., 2024). The SQL query retrieves the top three accounts by savings balance, and the Jinja2 template renders the results into natural language.

```
SQL Queries:
  - SELECT a."name", s."balance"
    FROM "ACCOUNTS" a
    JOIN "SAVINGS" s
      ON CAST(a."custid" AS DOUBLE) = s."custid"
    ORDER BY s."balance" DESC, a."name" ASC
    LIMIT 3;
Jinja2 Template:
    {% if values and values|length > 0 %}
       The three accounts with the highest savings balances are:
       {% for row in values %}
          - {{ row["name"] }} with a savings balance of {{ row["balance"]
              }}.
       {% endfor %}
       Overall, these represent the top savers by balance in the dataset.
    {% else %}
       No results were found for the requested top savings accounts.
    {% endif %}
```

improve accuracy but sacrifice scalability and privacy; and plan-based methods, such as TaPERA and SPaGe, yield only partially reusable plans. FACTS is the only approach satisfying all four desired properties.

## 3 METHODOLOGY

To avoid ambiguity, we first clarify the terminology used in this section. A user query denotes the natural language input provided by the user, which specifies an information need over one or more tables and may include rich contextual details. An SQL query refers to executable code generated by our method to retrieve the information required to satisfy the user query. A Jinja2 template is a rendering program that verbalizes SQL outputs into natural language. An offline template is the composite artifact introduced in this work, bundling one or more SQL queries together with a Jinja2 template. Unless otherwise specified, the term schema refers to the structural metadata of the table, e.g., column names and data types, rather than raw values. Finally, a summary denotes the final natural language output returned to the user after executing the SQL queries and rendering the Jinja2 template. The remainder of this section is structured as follows: Section 3.1 introduces the concept of offline templates and motivates their reusability; Section 3.2 details the LLM Council, which provides iterative validation and feedback; and Section 3.3 presents the complete FACTS framework and its three interconnected modules.

### 3.1 OFFLINE TEMPLATE

Formally, an offline template is defined as a composite artifact consisting of (1) one or more schema-aware SQL queries that retrieve relevant facts from the underlying tables, and (2) a Jinja2 template that transforms the retrieved outputs into a natural language summary. Crucially, offline templates are bound to both the table schema and the user query semantics. Once generated, the same offline template can be directly applied to any table sharing the same schema and answering the same user query or semantically similar queries, enabling reusability across tables that differ only in values, e.g., multiple years of financial records or multiple patients' health records. In this work, we define template reusability under an identical schema, without considering schema drift or renamed columns. This design avoids repeated LLM inference, provides efficiency through lightweight SQL execution, and ensures privacy compliance by never exposing raw table values to LLMs. Example 1 illustrates a real template generated by FACTS on the QFMTS dataset (Zhang et al., 2024). Here, the SQL query selects the top three accounts by savings balance from the ACCOUNTS and SAVINGS tables, and the Jinja2 template verbalizes the results into a coherent narrative. This example demonstrates offline

---

**Algorithm 1** LLM Council: Evaluate-and-Refine

---

**Input:** artifact $A$ (e.g., a guided question, SQL query, Jinja2 template, or summary), context $\mathcal{X}$ (schema, guidance, execution logs), model set $\mathcal{C} = \{M_1, \ldots, M_m\}$
**Output:** decision $\text{DEC} \in \{\texttt{YES}, \texttt{NO}\}$, consensus feedback FB
1: $\mathcal{R} \leftarrow \emptyset$                                               ▷ per-model results
2: **for** $M \in \mathcal{C}$ **do**                                  ▷ independent judgments
3:     $p \leftarrow \textsc{BuildPrompt}(A, \mathcal{X})$
4:     $o \leftarrow \textsc{LLMCall}(M, p)$                             ▷ LLM call
5:     $(d, f) \leftarrow \textsc{Parse}(o)$             ▷ $d \in \{\texttt{YES}, \texttt{NO}\}$, $f$=brief feedback
6:     $\mathcal{R} \leftarrow \mathcal{R} \cup \{(d, f)\}$
7: $\text{DEC} \leftarrow \textsc{MajorityVote}(\{d : (d, f) \in \mathcal{R}\})$
8: $\text{FB} \leftarrow \textsc{Aggregate}(\{f : (d, f) \in \mathcal{R}\}, \text{DEC})$         ▷ short consensus rationale
9: **return** $(\text{DEC}, \text{FB})$

---

templates are executable and reusable artifacts that faithfully capture user intent and generalize across tables with the same schema and query semantics.

### 3.2 LLM COUNCIL

The LLM Council is an ensemble of LLMs that collaboratively validate intermediate outputs at each stage of the FACTS framework. Rather than relying on a single model, the Council prompts multiple heterogeneous LLMs, each of which independently produces a structured judgment (`YES`/`NO`) and brief feedback. A majority-voting scheme determines whether a candidate artifact is accepted, while the collected feedback is aggregated into a consensus explanation that guides iterative refinement. The Council provides feedback in four places: (1) evaluating guided questions and filtering rules, (2) validating generated SQL queries, (3) checking alignment between SQL results and Jinja2 templates, and (4) assessing whether the final summary satisfies the user query. These validation steps will be described in detail in the next subsection. This mechanism reduces reliance on any single model, mitigates hallucinations, and ensures correctness and usability of generated artifacts.

Algorithm 1 presents the Council's evaluate-and-refine procedure in pseudocode. For each candidate artifact, a task-specific prompt is built from the artifact and its context, e.g., table schema, guided questions, execution logs, and passed independently to every model in the ensemble. Their responses are parsed into decisions and feedback, after which majority voting determines the overall acceptance decision, and aggregated feedback provides a concise rationale to guide refinement. Full prompt templates are included in Appendix A.1.

### 3.3 FACTS FRAMEWORK

The FACTS framework is composed of three interconnected stages, shown in Figure 2, with full pseudocode provided in Appendix A.2. At each stage, outputs generated by the LLM agent are validated by the LLM Council introduced in Section 3.2, which provides structured feedback and guides iterative refinement.

**Stage 1: Schema-Guided Specification and Filtering.** Given the user query and table schema, the agent first generates schema-aware clarifications in two complementary forms: (i) guided questions that identify which columns, relationships, and operations are relevant, and (ii) filtering rules that specify which rows or categorical values should be excluded. Crucially, the LLM never accesses the raw table contents. Instead, it proposes filtering rules in abstract form, e.g., "exclude rows where `category='expense'`", which are later expressed as `WHERE` clauses in SQL. This ensures the filtering process remains privacy-compliant and syntactically verifiable. For example, in the financial scenario introduced earlier, the agent may generate rules that remove irrelevant transaction categories, e.g., "exclude expense transactions", before producing summaries of gross income. The resulting schema-guided specifications serve as input to SQL synthesis in the next stage.

**Stage 2: SQL Queries Generation.** Using the approved specifications from Stage 1, the agent synthesizes one or more candidate SQL queries. These SQL queries integrate the filtering rules as

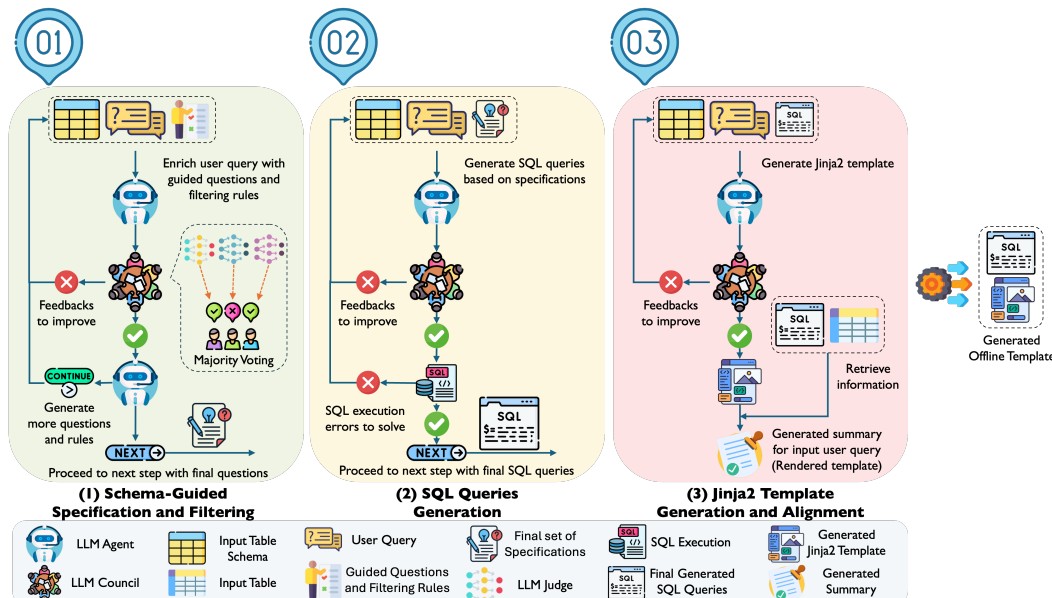

Figure 2: The FACTS framework for query-focused table summarization via Offline Template Generation. (1) **Schema-Guided Specification and Filtering**: the agent enriches the user query with guided questions and filtering rules over the table schema, with validation from the LLM Council. (2) **SQL Queries Generation**: using these specifications, the agent synthesizes and iteratively improves SQL queries through execution feedback and Council validation. (3) **Jinja2 Template Generation and Alignment**: a Jinja2 template verbalizes SQL outputs into natural language, with LLM Council checks ensuring alignment. The final output is a reusable offline template that combines validated SQL queries with a Jinja2 template.

constraints, ensuring that only relevant subsets of the data are processed. Each query is executed locally against the relevant tables to verify correctness. If a query fails or returns empty results, the error traces and execution outputs are passed to the LLM Council for feedback. Based on this feedback, the agent revises the query iteratively until it is executable. This refinement loop ensures that the final SQL queries are robust, accurate, and faithfully grounded in the user specification.

**Stage 3: Jinja2 Template Generation and Alignment.** Once the SQL queries are validated, the agent produces a Jinja2 template to render the results into natural language. The template is required to reference exact column names, correctly iterate over the returned rows, and handle empty results gracefully. The LLM Council then checks for alignment between SQL outputs and template references. If mismatches occur, e.g., missing fields or shape incompatibilities, the SQL and template are refined together until a consistent and valid pair is obtained. The final output is an offline template, consisting of reusable SQL queries and a Jinja2 template that can generalize across new tables with the same schema and query semantics.

Together, these three stages ensure FACTS achieves its key desired properties. Offline templates provide fast summarization by reusing validated SQL queries and Jinja2 template rendering logic, accurate outputs by grounding summaries in executed SQL queries rather than free-form generation, and privacy-compliant operation by exposing only schemas, without revealing raw table values. For completeness, the full prompts used in the FACTS framework are provided in Appendix A.3.

## 4 EXPERIMENTAL RESULTS

In this section, we present a comprehensive evaluation of the proposed FACTS framework. Our experiments are designed to address the following research questions: **RQ1:** Does FACTS, through offline templates, outperform existing methods for query-focused table summarization? **RQ2:** How does FACTS compare with non-agentic alternatives, such as directly prompting an LLM to generate an offline template in a single step? **RQ3:** To what extent does FACTS provide practical benefits in reusability and scalability, particularly when the schema and user query remain fixed or semantically

similar, or when table sizes increase? **RQ4:** Do human evaluators confirm that FACTS produces more factually correct and complete summaries with fewer hallucinations than existing baselines?

To answer these questions, we first introduce the datasets, evaluation metrics in Section 4.1, and baseline methods in Section 4.2. We then provide implementation details for FACTS and all baselines in Section 4.3, present the main results and analysis in Section 4.4, conduct ablation studies contrasting agentic versus single-call template generation in Section 4.5, and finally evaluate reusability and scalability in Section 4.6.

## 4.1 DATASET AND EVALUATION

**Datasets.** We evaluate FACTS on the test splits of three widely used benchmarks: FeTaQA (Nan et al., 2022), QTSumm (Zhao et al., 2023), and QFMTS (Zhang et al., 2024). FeTaQA consists of 2,003 examples from Wikipedia, each pairing a single relational table with a query and a short factual summary. QTSumm, also derived from Wikipedia, includes 1,078 examples where queries are linked to single tables but require generating longer, paragraph-style summaries. QFMTS contains 608 examples, with each query associated with an average of 1.8 tables, demanding reasoning and integration across multiple table schemas. QFMTS is based on the Spider dataset (Yu et al., 2018), which includes 200 databases spanning 138 distinct domains, such as university courses, online SQL tutorials, textbook examples, and public CSV repositories. Together, these datasets provide a complementary testbed: FeTaQA evaluates concise summarization, QTSumm emphasizes extended narrative responses, and QFMTS challenges systems with compositional multi-table reasoning.

**Evaluation Metrics.** We assess summarization quality using three standard automatic metrics. BLEU (Papineni et al., 2002) measures n-gram precision by computing exact word overlap between generated and reference summaries; we report SacreBLEU scores. ROUGE-L (Lin & Hovy, 2003) evaluates recall via the longest common subsequence, indicating how much reference content is covered; we report the F1 variant. METEOR (Banerjee & Lavie, 2005) balances precision and recall by considering unigram matches with stemming and synonymy. Together, these metrics provide a comprehensive assessment of both fluency and factual alignment in generated summaries.

## 4.2 BASELINE METHODS

We restrict our comparisons to training-free and fine-tuning-free approaches, since FACTS itself does not rely on supervised model adaptation. The baselines fall into two categories: prompt-based models and agentic frameworks. Prompt-based models include: **(1) Chain-of-Thought (CoT)** (Wei et al., 2022) prompts the LLM to explicitly verbalize intermediate reasoning steps before producing the final summary. **(2) DirectSumm** (Zhang et al., 2024) generates summaries in a single pass, conditioning directly on the table and user query. **(3) ReFactor** (Zhao et al., 2023) extracts query-relevant facts from the table and concatenates them with the user query as augmented input to the LLM. **(4) Reason-then-Summ** (Zhang et al., 2024) decomposes the process into two stages: first retrieving relevant facts, then composing a longer narrative summary. Agentic frameworks include: **(5) Binder** (Cheng et al., 2023) translates the query into executable SQL programs to ground the results in computation. **(6) Dater** (Ye et al., 2023) decomposes large tables into smaller ones and complex queries into simpler sub-queries, executes them individually, and aggregates their outputs. **(7) TaPERA** (Zhao et al., 2024) generates natural language plans that are converted into Python programs for execution and aggregation. **(8) SPaGe** (Zhang et al., 2025) introduces structured graph-based plans, improving reliability in multi-table scenarios. Together, these baselines cover the spectrum of training-free methods: (i) direct prompting of LLMs with or without explicit reasoning, and (ii) agentic approaches that couple LLMs with external executors.

## 4.3 IMPLEMENTATION DETAILS

The main LLM agent employs GPT-4o-mini as the backbone model, chosen for its strong performance in table reasoning and summarization tasks (Nguyen et al., 2025; Zhang et al., 2025). To further align outputs with target writing style, we employ in-context learning (Brown et al., 2020): when generating Jinja2 templates, the prompt includes three demonstration examples drawn from the corpus, encouraging summaries that are stylistically and structurally consistent with reference outputs.

At the workflow level, we allow a fixed upper bound of 10 guided questions and filtering rules, and set the patience for revision at 3 iterations for guided questions, filtering rules, SQL queries, and Jinja2 templates. SQL execution is handled by DuckDB (Raasveldt & Mühleisen, 2019), which

Table 2: Evaluations on the test sets of three benchmarks. FeTaQA and QTSumm are single-table datasets, while QFMTS is a multi-table dataset. The best and second-best results are shown in **bold** and underline, respectively. FACTS achieves the best or the second-best results on all datasets.

| Method | FeTaQA | | | QTSumm | | | QFMTS | | |
|---|---|---|---|---|---|---|---|---|---|
| | **BLEU** | **ROUGE-L** | **METEOR** | **BLEU** | **ROUGE-L** | **METEOR** | **BLEU** | **ROUGE-L** | **METEOR** |
| CoT | 28.2 | 51.0 | 56.9 | 19.3 | 39.0 | 47.2 | 31.5 | 54.3 | 58.1 |
| ReFactor | 26.2 | 53.6 | 57.2 | 19.9 | 39.5 | 48.8 | - | - | - |
| DirectSumm | 29.8 | 51.7 | 58.2 | 20.7 | 40.2 | 50.3 | 33.6 | 57.0 | 62.8 |
| Reason-then-Summ | 31.7 | 52.6 | 60.7 | 21.8 | 42.3 | **51.5** | 40.8 | 62.7 | 66.2 |
| Binder | 25.5 | 47.9 | 51.1 | 18.2 | 40.0 | 39.0 | 42.5 | 65.3 | 70.7 |
| Dater | 29.8 | 54.0 | 59.4 | 16.6 | 35.2 | 35.5 | - | - | - |
| TaPERA | 29.5 | 53.4 | 58.2 | 14.6 | 33.0 | 33.2 | - | - | - |
| SPaGe | **33.8** | 55.7 | 62.3 | 20.9 | 41.3 | 47.7 | 45.7 | 68.3 | **73.4** |
| **FACTS (GPT-Only) (ours)** | 30.8 | 55.7 | 66.0 | 20.1 | 43.1 | 50.5 | 45.4 | 70.5 | 73.2 |
| **FACTS (ours)** | 32.6 | **58.9** | **67.7** | 21.9 | **45.8** | 51.3 | **46.0** | **70.8** | 73.2 |

enables efficient in-memory querying and integrates seamlessly with pandas DataFrames in Python. The LLM Council consists of GPT-4o-mini, Claude-4 Sonnet, and DeepSeek v3. To further isolate the impact of Council composition, we also evaluate a **FACTS (GPT-Only)** variant, in which all three models in the Council are replaced with GPT-4o-mini, enabling us to assess the effectiveness of FACTS independent of cross-model diversity.

For baseline methods and other hyperparameters, we follow the setup of Zhang et al. (2025). All prompt-based and agentic baselines are implemented using the same GPT-4o-mini backbone to ensure comparability, and we directly cite reported results from Zhang et al. (2025) where available.

## 4.4 RESULTS AND ANALYSIS

**Effectiveness.** Table 2 reports results on the test splits of FeTaQA, QTSumm, and QFMTS. Overall, FACTS consistently achieves the best or second-best performance across all datasets and metrics, demonstrating the effectiveness of offline template generation with iterative validation. When compared with prompt-based methods, FACTS outperforms CoT, ReFactor, and DirectSumm across most metrics. These approaches lack grounding in executable programs, which makes them prone to hallucinations and incomplete coverage. Reason-then-Summ achieves relatively strong results on QTSumm, showing that explicitly structuring the generation process into fact retrieval and composition can sometimes improve the quality of the generated summaries. However, its gains are inconsistent across datasets, and like other prompt-based models, it lacks execution-level validation and remains vulnerable to factual errors and hallucinations in intermediate steps that may propagate into the final summary. Against agentic frameworks, FACTS surpasses Binder, Dater, and TaPERA, which often struggle with complex logic or multi-table reasoning. SPaGe remains a strong competitor by leveraging graph-based planning. Nevertheless, FACTS outperforms SPaGe on every dataset in at least two of the reported metrics, suggesting that FACTS generates more faithful and well-formed summaries that are better aligned with reference outputs. We further compare the full FACTS system with its GPT-Only variant, in which all three models in the LLM Council are replaced by GPT-4o-mini. While the full FACTS framework achieves the best overall performance, which we attribute in part to the diversity of reasoning behaviors introduced by heterogeneous Council members, the GPT-Only variant remains competitive and still outperforms baseline methods on most datasets and metrics. This result demonstrates that the core FACTS workflow itself is effective even without cross-model diversity, and that Council heterogeneity further amplifies these strengths.

**Computation Cost.** Across the three datasets evaluated, each sample involves on average 2.47 accepted guiding questions or filtering rules (2.25 initially accepted and 0.22 accepted after one round of revision), 1.36 SQL refinement rounds, and 1.84 template refinement rounds, with the maximum patience set to three rounds in our experiments. We further provide token-level efficiency analysis showing that the entire FACTS workflow requires $9,922$ input tokens and $1,045$ output tokens per sample on average (including all stages and Council outputs), offering a comprehensive view of runtime and generation cost.

Taken together, these findings provide a clear answer to **RQ1**, showing that FACTS reliably produces offline templates that deliver strong and stable performance across both single-table and multi-table summarization tasks, benefiting from the three interconnected stages introduced in Section 3.3. For additional qualitative insight, Appendix A.4 provides a step-by-step case study of FACTS, including intermediate outputs at each stage.

Table 3: Evaluations of Single-Call on the test sets of three benchmarks.

| Method | FeTaQA | | | QTSumm | | | QFMTS | | | Pass Rate |
|---|---|---|---|---|---|---|---|---|---|---|
| | **BLEU** | **ROUGE-L** | **METEOR** | **BLEU** | **ROUGE-L** | **METEOR** | **BLEU** | **ROUGE-L** | **METEOR** | |
| Single-Call | 29.4 | 52.1 | 58.4 | 14.2 | 37.9 | 40.6 | 35.4 | 63.2 | 69.8 | 83.2% |
| **FACTS (ours)** | **32.6** | **58.9** | **67.7** | **21.9** | **45.8** | **51.3** | **46.0** | **70.8** | **73.2** | **100.0%** |

Figure 3: Reusability analysis. Runtime for generating summaries with 1 versus 100 tables under the same schema and query.

Figure 4: Scalability analysis. Runtime for generating summaries as the number of rows in each table increases.

### 4.5 ABLATION STUDY

To better assess the role of iterative refinement, we compare FACTS with a simplified *Single-Call* variant. In this setting, the user query, table schema, and three in-context demonstration examples, identical to those used in our main experiments, are provided to GPT-4o-mini, which is prompted to generate an entire offline template in a single step, including both SQL queries and the Jinja2 template. Unlike FACTS, this approach does not incorporate iterative validation or feedback from the LLM Council, nor does it leverage local SQL execution traces during refinement. To capture robustness, we report the SQL *pass rate*, defined as the proportion of generated SQL queries that execute successfully without error.

Results are shown in Table 3. While Single-Call attains moderate text-level scores, it suffers from a substantially lower pass rate. This illustrates the brittleness of one-shot template generation: SQL queries often contain syntax errors, reference non-existent columns, or yield empty outputs, which directly undermines summary quality. By contrast, FACTS consistently achieves a 100% pass rate across datasets, as its iterative refinement loop with Council validation and execution feedback detects and corrects errors before finalization. This not only ensures robustness but also translates into consistently higher BLEU, ROUGE-L, and METEOR scores. In summary, these results directly address **RQ2**, confirming that FACTS substantially outperforms non-agentic single-step alternatives by combining structured stages with iterative validation.

### 4.6 REUSABILITY AND SCALABILITY ANALYSIS

Finally, we examine whether FACTS delivers the promised advantages of reusability and scalability, addressing **RQ3**. We compare against two strong baselines, Reason-then-Summ and SPaGe, using 100 randomly sampled examples from QTSumm.

**Experiment 1: Reusability across tables.** We fix the user query and table schema but vary the cell values. As shown in Figure 3, with a single table, FACTS is slightly slower than Reason-then-Summ, since it must generate the offline template for the first time, and comparable to SPaGe. We emphasize that the latency reported in Figure 3 *already includes* the cost of the initial offline template generation. However, once the template is generated, FACTS achieves a substantial speed advantage when reusing it across multiple tables with the same schema. With 100 tables under the same schema, FACTS dramatically outperforms both baselines: new summaries require only SQL execution and Jinja2 rendering, while the other methods must reprocess the entire table for every example.

**Experiment 2: Reusability across semantically similar queries.** To assess robustness to semantically similar user queries, we conduct an additional experiment on the same 100 randomly sampled QTSumm examples. We use GPT-5 to paraphrase both the user queries and corresponding refer-

ence summaries, creating semantically similar but lexically different variants. The original offline templates are then applied without regeneration. FACTS maintains comparable performance with BLEU = 21.8, ROUGE-L = 43.5, and METEOR = 50.8, confirming that it generalizes robustly to semantic variations of the same query while preserving effectiveness.

**Experiment 3: Scalability with table size.** We next test how runtime scales as the number of rows in each table increases, ranging from 100 to 1000. Figure 4 shows that FACTS remains flat in runtime, as templates depend only on the schema. By contrast, Reason-then-Summ and SPaGe incur steadily increasing cost, since larger tables must be serialized and passed into the LLM.

Together, these results show that FACTS achieves both reusability and scalability, while preserving summary quality. This combination of speed, reliability, and accuracy makes it particularly well-suited for real-world deployments.

### 4.7 HUMAN EVALUATION AND HUMAN PREFERENCE STUDY

To complement the automatic and computational evaluations, we perform human assessments.

**Human Evaluation.** We conduct a comprehensive human evaluation on 100 randomly sampled examples from QTSumm and 100 from QFMTS to assess four aspects: (1) whether each generated SQL semantically matches the user query (intent match), (2) whether the SQL execution results correctly correspond to the numerical or factual content in the reference summary (SQL execution accuracy), (3) whether the numbers and facts rendered in the final summary faithfully reflect the SQL execution results (template rendering accuracy), and (4) whether the LLM Council unanimously accepts a specification or SQL query that leads to an incorrect result (Council consensus error rate). FACTS achieves 97% intent match, 94% SQL execution accuracy, and 98% template rendering accuracy, with a very low Council consensus error rate of about 3%. The Council consensus error rate is computed across two stages: (i) schema-guided specification and filtering, and (ii) SQL query generation. While no errors occur during the specification stage, about 6% of the SQL queries approved by the Council lead to incorrect results during the generation stage, yielding an overall average error rate of approximately 3%. The SQL execution accuracy is lower than the template rendering accuracy because some SQL queries compute incorrect values with respect to the reference summary, while the rendering accuracy reflects whether the template correctly verbalizes the SQL execution results. Therefore, the overall factual correctness of the generated summaries can be estimated as $94\% \times 98\% \approx 92\%$. Although FACTS achieves high factual accuracy, the automatic metrics primarily capture surface-level overlap and semantic similarity rather than factual correctness. Human evaluation thus provides complementary insights beyond what automatic metrics can measure.

**Human Preference Study.** We further conduct a side-by-side human preference study comparing FACTS with the strongest baseline SPaGe on QFMTS. Human evaluators are presented with the same user query and two randomly ordered system outputs, one from FACTS and one from SPaGe, without method identifiers. Evaluators are then asked to choose the preferred output or indicate no preference based on three criteria: (1) whether the summary fully answers the user query (completeness), (2) whether the reported numbers and facts are accurate (correctness), and (3) whether unsupported or ungrounded content is introduced (hallucination). FACTS is preferred in 55% of cases for completeness, 59% for correctness, and 60% for hallucination reduction, indicating that human evaluators consistently favor FACTS for producing more accurate, complete, and faithful summaries. These findings confirm that human judgments align with the automatic metrics and computational analyses, collectively addressing **RQ4**.

## 5 CONCLUSION

In this work, we address the challenges in query-focused table summarization by proposing **FACTS**, an agentic framework that generates reusable offline templates by combining schema-guided specifications, SQL synthesis, and Jinja2 rendering, with iterative validation from an LLM Council. Extensive experiments show that FACTS consistently outperforms strong baselines, while offering unique advantages in reusability, scalability, and privacy compliance. We also acknowledge that FACTS assumes a practical privacy model where only table schemas and queries are shared with external LLMs, while raw values remain local. A detailed discussion of this privacy scope, limitations, and possible extensions is provided in Appendix A.5. These results highlight FACTS as a practical solution for query-focused table summarization in real-world applications.

## REPRODUCIBILITY STATEMENT

For reproducibility, Section 4.3 outlines the implementation details of our method, while Appendix A.1 and Appendix A.3 provides the detailed prompts used in our method. We will publicly release the complete codebase once this paper is accepted.

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

LLM USAGE

We made use of large language models solely for improving the presentation of this paper. Their role was limited to refining wording and verifying grammar to enhance clarity and readability. No assistance from LLMs was involved in the design of methods, implementation of experiments, or analysis of results.

# A APPENDIX

## A.1 LLM COUNCIL PROMPTS

For completeness, we include the full prompts used by the LLM Council for evaluation. These prompts directly correspond to the four validation steps described in Section 3.2 and formalized in Algorithm 1. Each prompt is presented independently to all LLMs in the Council, and their structured outputs are aggregated via majority voting with feedback consolidation. We provide the exact versions here to ensure transparency and reproducibility of our experiments.

Example 2: Prompt for evaluating guided questions and filtering rules.

```
You are evaluating a question or filtering rule for table summarization.

Table Information:
[table schema here]

User Query:
[original user query]

Previously Generated Questions or Filtering Rules:
[list of previously accepted guided questions and filtering rules]

Current Question or Filtering Rule to Evaluate:
[proposed guiding question or filtering rule]

Is this a good question or filtering rule that will help guide SQL query
    generation? Answer with YES or NO only.

If NO, provide a brief reason why this question is not helpful.

Output format:
Decision: [YES/NO]
Feedback: [Brief reason if NO, or 'Question is good' if YES]
```

Example 3: Prompt for evaluating SQL queries.

```
You are evaluating a SQL query execution for table summarization.

Table Information:
[table schema here]

Guidance:
[generated guided questions and filtering rules]

SQL Query:
[proposed SQL query]

Execution Result:
[empty results or error message]

Evaluate whether this SQL query is valid and appropriate:
1. Does it execute without errors?
2. Does it return the non-empty data for summarization?
3. Does it filter and select appropriate columns?

Answer with YES or NO only. If NO, provide a brief reason.

Output format:
Decision: [YES/NO]
Feedback: [Brief reason if NO, or 'SQL query is good' if YES]
```

Example 4: Prompt for evaluating SQL–template alignment.

```
You are evaluating whether a SQL query result aligns with a Jinja2
    template for table summarization.

Table Information:
[table schema here]

SQL Query:
[proposed SQL query]

Jinja2 Template:
[proposed Jinja2 template]

Evaluate:
1. Does the SQL return all fields that the template tries to access?
2. Is the data structure compatible (e.g., if template expects multiple
    rows, does SQL return them)?
3. Are field names in the template matching the column names returned by
    SQL?

Answer with YES or NO only. If NO, provide a brief reason.

Output format:
Decision: [YES/NO]
Feedback: [Brief reason if NO, or 'SQL and template are well-aligned' if
    YES]
```

Example 5: Prompt for evaluating generated summaries.

```
You are evaluating a generated summary for table summarization.

Table Information:
[table schema here]

User Query:
[original user query]

Generated Summary:
[system-produced summary]

Evaluate summary quality:
1. Relevance to the query
2. Accuracy of information
3. Clarity and coherence
4. Completeness

Answer with YES or NO only. If NO, provide a brief reason.

Output format:
Decision: [YES/NO]
Feedback: [Brief reason if NO, or 'Summary is good' if YES]
```

## A.2  PSEUDO CODE OF FACTS

Algorithm 2 summarizes the FACTS workflow. The process begins with **Schema-Guided Specification and Filtering**, where the agent proposes schema-aware clarifying questions and filtering rules based on the user query $q$ and table schema $\mathcal{S}$. Each specification is vetted by the LLM Council, and accepted ones are accumulated in $\mathcal{U}$ to progressively refine the query intent. Next, in **SQL Queries Generation**, candidate SQL queries are synthesized using $(q, \mathcal{U}, \mathcal{S})$, executed locally against the table, and validated both by execution feedback and the LLM Council. Invalid queries are iteratively revised until a correct and executable query $\mathcal{Q}$ is obtained. Finally, in **Jinja2 Template Generation and Alignment**, the agent generates a Jinja2 template $\mathcal{J}$ to render SQL results into natural language. The LLM Council checks alignment between template references and SQL outputs; if misalignments are detected, the template is refined until valid. The resulting offline template $\mathcal{T} = (\mathcal{Q}, \mathcal{J})$ can then be reused across any table with the same schema, enabling fast, accurate, and privacy-compliant summarization without exposing raw table values to the LLMs.

## A.3  FACTS PROMPTS

FACTS relies on a set of carefully designed prompts to guide schema-aware question and filtering rule generation, SQL synthesis, and Jinja2 template construction. Below we provide a few representative examples; the complete set of prompts, including dataset-specific variants due to multi-table schemas, is released with our code for reproducibility.

---

**Algorithm 2** FACTS Framework

---

**Input:** user query $q$, table schema $\mathcal{S}$, LLMs Council $\mathcal{C}$, max number of guiding specifications $K_q$, patience $P_q, P_{\text{sql}}, P_{\text{tpl}}$
**Output:** offline template $\mathcal{T} = (\mathcal{Q}, \mathcal{J})$ where $\mathcal{Q}$ is a set of SQL queries and $\mathcal{J}$ is a Jinja2 template
1: /* Component 1:  Schema-Guided Specification and Filtering */
2: $\mathcal{U} \leftarrow \emptyset$        ▷ accepted guiding spefications
3: **for** $k = 1$ **to** $K_q$ **do**        ▷ how many guiding specifications we can generate
4:     $u \leftarrow$ GENSPECIFICATION($q, \mathcal{S}, \mathcal{U}$)
5:     (vote, fb) $\leftarrow$ COUNCILJUDGE($\mathcal{C}, u$)
6:     **if** vote $=$ YES **then**
7:        $\mathcal{U} \leftarrow \mathcal{U} \cup \{u\}$        ▷ added to the accepted set of guiding specifications
8:     **else**
9:        $t \leftarrow 0$
10:        **while** vote $\neq$ YES **and** $t < P_q$ **do**        ▷ refine until Council satisfied or reach patience
11:           $u \leftarrow$ REVISESPECIFICATION($u, \text{fb}, q, \mathcal{S}, \mathcal{U}$)
12:           (vote, fb) $\leftarrow$ COUNCILJUDGE($\mathcal{C}, u$)
13:           $t \leftarrow t + 1$
14:        **if** vote $=$ YES **then**
15:           $\mathcal{U} \leftarrow \mathcal{U} \cup \{u\}$
16:     **if** SUFFICIENT($\mathcal{U}$) **then break**        ▷ final set of guiding specifications
17: /* Component 2:  SQL Queries Generation */
18: $\mathcal{Q} \leftarrow \emptyset$; vote $\leftarrow$ false
19: $t \leftarrow 0$
20: **while not** vote **and** $t < P_{\text{sql}}$ **do**
21:     $\tilde{\mathcal{Q}} \leftarrow$ GENSQL($q, \mathcal{U}, \mathcal{S}$)
22:     exec $\leftarrow$ EXECUTESQL($\tilde{\mathcal{Q}}, \mathcal{S}$)
23:     (vote, fb) $\leftarrow$ COUNCILJUDGE($\mathcal{C}, (\tilde{\mathcal{Q}}, \text{exec})$)
24:     **if** vote $=$ YES **and** VALID(exec) **then**
25:        $\mathcal{Q} \leftarrow \tilde{\mathcal{Q}}$; vote $\leftarrow$ true
26:     **else**
27:        $\tilde{\mathcal{Q}} \leftarrow$ REVISESQL($\tilde{\mathcal{Q}}, \text{exec}, \text{fb}$)        ▷ handle errors, empties, shape mismatches
28:        $t \leftarrow t + 1$
29: /* Component 3:  Jinja2 Template Generation and Alignment */
30: vote $\leftarrow$ false; $t \leftarrow 0$
31: **while not** vote **and** $t < P_{\text{tpl}}$ **do**
32:     $\mathcal{J} \leftarrow$ GENJINJA2($q, \mathcal{Q}, \mathcal{S}$)
33:     (vote, fb) $\leftarrow$ COUNCILJUDGE($\mathcal{C}, (\mathcal{J}, \mathcal{Q}, \mathcal{S})$)
34:     vote $\leftarrow$ (vote $=$ YES) **and** ALIGNED($\mathcal{J}, \mathcal{Q}, \mathcal{S}$)        ▷ fields match SQL outputs
35:     **if not** vote **then**
36:        $(\mathcal{Q}, \mathcal{J}) \leftarrow$ REFINE($\mathcal{Q}, \mathcal{J}, \text{fb}$)        ▷ fix unknown fields, shapes
37:        $t \leftarrow t + 1$
38: **return** $\mathcal{T} = (\mathcal{Q}, \mathcal{J})$        ▷ reusable offline template

---

Example 6: Prompt for generating a schema-aware guiding question and filtering rule.

```
Based on the table information and user query below, generate ONE
    specific, detailed question or filtering rule that will help guide
    SQL query generation.

Table Information:
[table schema here]

User Query: [user query here]

Previously generated questions and filtering rules:
[None or list of prior questions and filtering rules]

Generate ONE new question or filtering rule that:
```

```
1. Is different from previously generated questions and filtering rules
2. Clarifies what specific information is needed or what information is
    irrelevant
3. Helps understand data relationships
4. Guides the SQL query structure

Output format:
Specification: [Your single question or filtering rule here]
```

Example 7: Prompt for SQL query synthesis.

```
Based on the table information, user query, and refined questions below,
    generate a valid DuckDB SQL query.

Table Information:
[table schema here]

Guided Specifications:
[final set of guided questions and filtering rules]

IMPORTANT: You are querying a pandas DataFrame named 'df' that contains
    the table data.

Generate valid DuckDB SQL SELECT query that:
1. Retrieves the necessary information to answer the user query
2. Uses proper DuckDB syntax
3. References the DataFrame as 'df'
4. Quotes column names exactly as they appear
5. Handles data types appropriately

Output format:
SQL queries:
[Your SQL query here]
```

Example 8: Prompt for Jinja2 template generation.

```
Based on the demonstration examples below and the current SQL result,
    generate a Jinja2 template.

--- Demonstration Examples ---
[table, user query, and reference summary triples]

--- Current Task ---
Table Information: [table schema here]
User Query: [user query here]
SQL Query: [SQL query here]

Generate a Jinja2 template that:
1. Uses the variable name 'values' to access the data
2. Iterates with {% for row in values %}
3. Accesses fields with row["Column Name"]
4. Produces a coherent paragraph summary in the style of the examples
5. Handles empty results gracefully

Output format:
Jinja2 template:
[Your Jinja2 template here]
```

For space reasons, we only show these representative prompts here. The full set, including iterative improvement and alignment prompts, is available in our code release.

## A.4    CASE STUDY: STEP-BY-STEP OUTPUTS ON QFMTS

We illustrate FACTS end-to-end on a QFMTS example, ID #303. The user asks: "Show all document names using templates with template type code BK." We show the intermediate artifacts produced at each stage.

Example 9: Input (QFMTS #303): user query and schemas.

```
User Query:
  Show all document names using templates with template type code BK.

Schemas:
  Templates(Template_ID, Version_Number, Template_Type_Code,
      Date_Effective_From, Date_Effective_To, Template_Details)
  Documents(Document_ID, Template_ID, Document_Name, Document_Description
      , Other_Details)
```

Example 10: Stage 1 — Accepted guided specifications.

```
Q1: Should results include only documents whose associated BK templates
    are currently effective
    (i.e., Date_Effective_From <= today and (Date_Effective_To is null or
        >= today)),
    or should all BK templates be considered regardless of dates?

Q2: Should we return all documents whose Template_ID links to a Templates
     row with Template_Type_Code = 'BK'
    regardless of the templates' effective date range, or limit results
        to BK templates that are currently effective based on
        Date_Effective_From and Date_Effective_To?
```

Example 11: Stage 2 — Synthesized SQL (validated by Council).

```
SELECT d."Document_Name"
FROM "Documents" AS d
JOIN "Templates" AS t
  ON d."Template_ID" = t."Template_ID"
WHERE t."Template_Type_Code" = 'BK';
```

Example 12: Stage 3 — Final Jinja2 template (after refinement).

```
{% set names = (values | map(attribute='Document_Name') | select() | list
    ) %}
{% set unique_names = names | unique | list %}
{% if unique_names and unique_names|length > 0 %}
There are {{ unique_names|length }} documents that use templates with the
    template type code BK.
The document names are {% for n in unique_names %}{{ n }}{% if not loop.
    last %}{% if loop.revindex == 1 %}, and {% else %}, {% endif %}{%
    endif %}{% endfor %}.
{% else %}
There are 0 documents that use templates with the template type code BK.
{% endif %}
```

Example 13: Final output vs. reference.

```
Generated summary:
  There are 5 documents that use templates with the template type code BK
      . The document names are Robbin CV, Data base, How to read a book,
      Palm reading, About Korea.

Reference summary:
  There are 5 document names that use templates with the template type
      code BK. The document names are Robbin CV, Data base, How to read a
      book, Palm reading, and About Korea.

Single-example scores:
  SacreBLEU = 83.2, ROUGE-L = 93.5, METEOR = 95.2
```

**Summary.** The accepted guided specifications focus the retrieval criterion, the synthesized SQL grounds the result set, and the refined Jinja2 template ensures correct counting and list formatting. The final output faithfully matches the reference.

## A.5 PRIVACY SCOPE AND THREAT MODEL

To clarify the privacy assumptions of FACTS, we adopt a practical enterprise-level threat model in which the large language model (LLM) is treated as an external API that can observe schema-level prompts but cannot access any local data or SQL execution results. Raw table cell values (e.g., personal identifiers, transaction records, or numerical measurements) are considered sensitive and never leave the local environment. All interactions with LLMs occur solely at the schema or query level during guided-question generation, SQL synthesis, and template rendering, while SQL execution and summary rendering are performed locally. This design ensures that only structural information, not actual data, is exposed. We acknowledge that schema structures or user queries may still reveal limited information about domain or intent. Future work may integrate stronger defenses such as schema abstraction, name obfuscation, or query redaction to further strengthen privacy protection.

