# OpenReview forum: "FACTS: Table Summarization via Offline Template Generation with Agentic Workflows"
_ICLR.cc/2026/Conference — ICLR 2026 Conference Withdrawn Submission_

### Official Review · Reviewer_xc11 · 2025-11-01

**Soundness:** 2
**Presentation:** 3
**Contribution:** 2
**Rating:** 4
**Confidence:** 2

**Summary:**

This paper introduces FACTS, a query-focused table summarization framework that generates reusable offline templates composed of schema-aware SQL queries and a Jinja2 text renderer. The system is designed to be fast, accurate, and privacy-compliant by sending only table schemas—not cell values—to language models. FACTS follows a three-stage agentic workflow: it first elicits schema-guided clarifications and filtering rules, then synthesizes and validates executable SQL, and finally produces a Jinja2 template aligned with the SQL outputs. An LLM Council—an ensemble that votes and provides feedback—audits artifacts at every stage to improve correctness and robustness. According to the workflow diagram on page 6, this process yields a template that can be applied across any table sharing the same schema and query semantics, enabling amortized speedups. The example on page 4 shows how the approach selects top savers via SQL and renders a narrative via Jinja2, while the comparison diagram on page 2 emphasizes gains in reusability and privacy over direct prompting. Experiments on FeTaQA, QTSumm, and QFMTS report best or second-best scores across BLEU, ROUGE-L, and METEOR (page 8), a 100% SQL pass rate versus an 83.2% single-call baseline (page 9), and substantial runtime advantages when reusing templates across many tables or as table size grows (page 9). The appendix provides prompts, pseudocode, and a step-by-step case study that illustrate the system’s mechanics (pages 14–19).

**Strengths:**

The work is original in framing query-focused table summarization as offline template generation that deliberately separates computation (SQL) from surface realization (Jinja2) and binds both to the table schema. This abstraction moves beyond natural-language plans and ad-hoc program synthesis by yielding a concrete, reusable artifact that amortizes LLM cost and avoids re-prompting for recurring queries, which is convincingly demonstrated by the reusability and scalability plots on page 9.

Methodological quality is good: the three-stage agentic pipeline with Council-style validation reduces brittle one-shot failures, achieves a 100% SQL pass rate relative to a single-call baseline (page 9), and grounds all textual claims in executed queries, improving factuality. Clarity is high: the paper defines terms precisely, presents a clear end-to-end workflow (page 6), gives a concrete running example (page 4), documents prompts and pseudocode (pages 14–18), and provides a case study that shows intermediate artifacts and the final rendered summary (pages 19–20).

**Weaknesses:**

The empirical evaluation relies exclusively on automatic overlap metrics, which only imperfectly capture factual faithfulness and utility; a human evaluation or auditor-based factuality check would strengthen claims about summary quality.

The privacy story, while thoughtful, could be more rigorous: although values are never sent to LLMs, schemas and query text can still leak sensitive structure or intent; articulating a formal threat model and the conditions under which schemas are safe would clarify limits. The LLM Council improves robustness, but its cost/latency and sensitivity to council composition, temperature, and prompt phrasing are not quantified; an ablation isolating each validation point would disentangle where the gains arise.

Generalization beyond DuckDB and portability across SQL dialects are not evaluated, yet production environments often involve diverse engines; similarly, the approach presumes stable, clean schemas, while many real datasets require entity resolution and schema drift handling.

The single-call baseline is a relatively weak foil for the agentic method; comparisons against stronger structured-program baselines configured for reusability would better calibrate the contribution.

while the runtime plots on page 9 are compelling, a full accounting that includes initial template-creation time, council iterations, and the overhead of guided-question generation would help practitioners plan deployments, and releasing code only upon acceptance limits current reproducibility.

**Questions:**

See weaknesses

---

> ### Author Response · Authors · 2025-11-21
>
> We sincerely thank Reviewer xc11 for the thoughtful and constructive feedback. We have carefully revised the manuscript in response to the reviewer’s comments, and all corresponding changes are highlighted in red in the revised PDF. Below, we provide point-by-point responses to the weaknesses and questions raised by the reviewer.
>
> ## Weakness
> > The empirical evaluation relies exclusively on automatic overlap metrics...
>
> We thank the reviewer for this valuable comment.
> We agree that automatic overlap metrics alone cannot fully capture factual faithfulness or practical utility.
> To address this limitation, we have added a comprehensive human evaluation in Section 4.7 (lines 502–529) of the revised manuscript.
> In this evaluation, we assess 200 randomly sampled examples (100 from QTSumm and 100 from QFMTS) across four aspects:
> (1) whether each generated SQL semantically matches the user query (intent match),
> (2) whether the SQL execution results correctly correspond to the numerical or factual content in the reference summary (SQL execution accuracy),
> (3) whether the numbers and facts rendered in the final summary faithfully reflect the SQL execution results (template rendering accuracy), and
> (4) whether the LLM Council unanimously accepts a specification or SQL query that leads to an incorrect result (Council consensus error rate).
> The Council consensus error rate is computed across two stages: (i) schema-guided specification and filtering, and (ii) SQL query generation.
> While no errors occur during the specification stage, about 6\% of the SQL queries approved by the Council lead to incorrect results during the generation stage, yielding an overall average error rate of 3\%.
> Overall, FACTS achieves 97\% intent match, 94\% SQL execution accuracy, 98\% template rendering accuracy, and 3\% Council consensus error rate, confirming strong factual reliability.
> We note that the SQL execution accuracy is lower than the template rendering accuracy because some SQL queries compute incorrect values with respect to the reference summary, while the rendering accuracy reflects whether the template correctly verbalizes the SQL execution results.
> Therefore, the overall factual correctness of the generated summaries can be estimated as $94\\% \times 98\\% \approx 92\\%$.
> In addition, we conduct a side-by-side human preference study comparing FACTS with the strongest baseline SPaGe, where annotators judge completeness, correctness, and hallucination.
> FACTS is preferred in 55\% of cases for completeness, 59\% for correctness, and 60\% for hallucination reduction.
> These results collectively demonstrate that FACTS produces factually accurate and high-quality summaries, validating its effectiveness beyond automatic metrics.
>
> > The privacy story, while thoughtful, could be more rigorous: although values are never sent to LLMs, schemas and query text can still leak sensitive structure or intent; articulating a formal threat model and the conditions under which schemas are safe would clarify limits.
>
> We thank the reviewer for this thoughtful suggestion.
> To make our privacy analysis more rigorous, we have clarified the privacy assumptions of FACTS and formally articulated the threat model in the revised manuscript.
> Specifically, we added the following statement in the Conclusion section (lines 536–539):
> "We also acknowledge that FACTS assumes a practical privacy model where only table schemas and queries are shared with external LLMs, while raw values remain local. A detailed discussion of this privacy scope, limitations, and possible extensions is provided in Appendix A.5."
> Appendix A.5 now provides a detailed description of our threat model, which specifies the adversary assumptions, sensitive data scope, and privacy guarantees:
> "FACTS adopts a practical enterprise-level threat model in which the LLM is treated as an external API that can observe schema-level prompts but cannot access any local data or SQL execution results. Raw table cell values (e.g., personal identifiers, transaction records, or numerical measurements) are considered sensitive and never leave the local environment. All interactions with LLMs occur solely at the schema or query level during guided-question generation, SQL synthesis, and template rendering, while SQL execution and summary rendering are performed locally. This design ensures that only structural information, not actual data, is exposed. We acknowledge that schema structures or user queries may still reveal limited information about domain or intent. Future work may integrate stronger defenses such as schema abstraction, name obfuscation, or query redaction to further strengthen privacy protection."
> These additions clearly define the privacy scope and assumptions of FACTS, addressing the reviewer’s concern by providing a rigorous and transparent threat model.

---

> ### Author Response · Authors · 2025-11-21
>
> > The LLM Council improves robustness, but its cost/latency and sensitivity to council composition, temperature, and prompt phrasing are not quantified; an ablation isolating each validation point would disentangle where the gains arise.
>
> We thank the reviewer for this valuable comment.
> Regarding cost and latency, the runtime reported in Figure 3 already includes the cost of the LLM Council, as template generation and validation are measured end-to-end.
> We emphasize this in line 478-479 in the revised version.
> In the revised manuscript, we further provide detailed quantitative analysis of computation cost in Section 4.4 (lines 421–427).
> On average, each sample involves 2.47 accepted guiding questions or filtering rules (2.25 initially accepted and 0.22 accepted after revision), 1.36 SQL refinement rounds, and 1.84 template refinement rounds (maximum = 3).
> We also report token-level efficiency statistics showing that the entire FACTS workflow requires approximately 9,922 input tokens and 1,045 output tokens per sample (including all stages and Council outputs), providing a comprehensive view of computational cost.
> These results demonstrate that the iterative Council validation introduces minimal overhead relative to the efficiency gains achieved through offline template reuse.
>
> For sensitivity to Council composition, we have added an ablation study in the revised manuscript (Section 4.3, lines 392–395, and Section 4.4, lines 414–420) with a **FACTS (GPT-Only)** variant, where all three Council models are replaced with GPT-4o-mini.
> The results are included in Table 2.
> This experiment shows that while the full heterogeneous Council achieves the best overall performance, partly due to the diversity of reasoning behaviors introduced by different LLMs, the GPT-Only variant remains competitive and still outperforms most baselines, confirming that the core FACTS workflow is intrinsically effective even without cross-model diversity.
>
> For temperature settings, we follow the same configuration as SPaGe and clearly note this in the implementation details (line 396):
> "For baseline methods and other hyperparameters, we follow the setup of Zhang et al. (2025)."
> We set the temperature and top-p to 0.1 and 0.95, respectively.
>
> In addtion, regarding prompt phrasing, we do not perform any fine-grained prompt engineering.
> Each stage uses a clear and standardized task description to specify the expected output.
> Because FACTS is an agentic multi-stage workflow, it is difficult to conduct controlled experiments that vary prompt wording in isolation without affecting other stages.
> We therefore view systematic exploration of prompt phrasing sensitivity, which primarily depends on the intrinsic robustness of the underlying LLMs, as an interesting direction for future work.
>
> Finally, we want to emphasize that, to better understand the source of performance gains, **our original paper** also includes an ablation study with a **Single-Call** variant (Section 4.5, Table 3).
> This ablated variant attains only moderate text-level scores and exhibits a substantially lower SQL pass rate, whereas FACTS consistently achieves a 100\% pass rate across all datasets.
> These results confirm that iterative validation and refinement are crucial to the robustness and overall quality of FACTS.

---

> ### Author Response · Authors · 2025-11-21
>
> > Generalization beyond DuckDB and portability across SQL dialects are not evaluated, yet production environments often involve diverse engines; similarly, the approach presumes stable, clean schemas, while many real datasets require entity resolution and schema drift handling.
>
> We appreciate the reviewer’s practical perspective on this question. We chose DuckDB primarily for its seamless integration with Python, which makes it well-suited for our agentic workflow involving iterative SQL execution and validation. The SQL queries generated by FACTS strictly follow standard SQL syntax. In our manuscript, we explicitly clarify that this choice is implementation-based in line 377-391, rather than a methodological limitation.
>
> Regarding the assumption of clean schemas, we agree that real-world raw data may involve entity resolution or schema drift issues before analysis. These preprocessing challenges are important, but our main focus is on efficient and privacy-compliant summarization given a structured schema. The benchmark datasets we use already include diverse and realistic database schemas, reflecting practical real-world scenarios.
>
> Regarding schema variations, our approach is robust to simple changes such as renamed columns, which can be handled by updating the column names in the generated SQL queries.
> For more substantial schema changes (e.g., added columns), our agentic workflow needs to be executed again to regenerate the offline templates.
> However, as shown in Figure3, the overall latency remains comparable to SPaGe even when the offline template must be generated from scratch.
> Moreover, once the new template is generated, it can be efficiently reused across all tables sharing the same schema and semantically similar queries.
> We consider handling more complex schema drift without regenerating templates beyond the current scope, but note that FACTS remains practical and competitive even when regeneration is required.
>
> > The single-call baseline is a relatively weak foil for the agentic method; comparisons against stronger structured-program baselines configured for reusability would better calibrate the contribution.
>
> We would like to thank the reviewer for the comment and clarify that the **Single-Call** setup is not used as a baseline but rather as an ablation study to isolate the effect of the iterative refinement process within our agentic workflow. The goal of this comparison is to demonstrate that iterative validation and correction by the LLM Council significantly improve the correctness of the generated SQL queries. For benchmarking against existing structured-program or agentic methods, our main experiments already include strong baselines such as Binder, Dater, TaPERA, and SPaGe. As shown in Table2 and Figures3–4, FACTS consistently achieves competitive or better performance than these baselines while providing additional advantages in reusability, scalability, and privacy compliance.
>
> > while the runtime plots on page 9 are compelling, a full accounting that includes initial template-creation time, council iterations, and the overhead of guided-question generation would help practitioners plan deployments
>
> We thank the reviewer for this practical suggestion.
> The runtime plots in Figure 3 already include the cost of initial offline template generation, Council iterations, and guided-question generation, as template creation and validation are measured end-to-end.
> In the revised manuscript, we explicitly clarify this in Section 4.6 (lines 478–479):
> "We emphasize that the latency reported in Figure 3 already includes the cost of the initial offline template generation."
> Furthermore, we provide a detailed analysis of computation cost in Section 4.4 (lines 421–427), reporting that each sample involves on average 2.47 accepted guiding questions or filtering rules (2.25 initially accepted and 0.22 after revision), 1.36 SQL refinement rounds, and 1.84 template refinement rounds (maximum = 3).
> We also include token-level efficiency analysis showing that the entire FACTS workflow requires approximately 9,922 input tokens and 1,045 output tokens per sample (including all stages and Council outputs), offering a comprehensive breakdown of runtime and generation cost.
> These additions demonstrate that the overall iterative and Council-related overhead is minimal relative to the substantial efficiency gains achieved through template reuse.

---

> ### Author Response · Authors · 2025-11-21
>
> > releasing code only upon acceptance limits current reproducibility.
>
> We appreciate the reviewer’s concern. Due to institutional policy, we are not permitted to release the code before the acceptance of this work. However, we have made every effort to ensure reproducibility. Section4.1 elaborates the details of datasets and evaluation metrics. Section4.3 of the paper provides necessary implementation details. The appendix further includes the full pseudocode of the FACTS workflow and all prompt templates used in each stage. Together, these materials allow our method to be independently reproduced. We will publicly release the complete codebase immediately upon acceptance in accordance with institutional policy.

---

> ### Author Response · Authors · 2025-11-24
> **Looking Forward to Your Feedback**
>
> Dear Reviewer xc11,
>
> We hope this message finds you well. We would like to kindly follow up regarding the rebuttal feedback for our submission. We truly appreciate the time and effort you have devoted to reviewing our paper, and we are happy to provide any clarifications if needed.
>
> We understand that this is a busy period and greatly value your consideration. Please let us know if any further information would be helpful from our side.
>
> With kind regards,
>
> Authors

---

> ### Author Response · Authors · 2025-11-28
> **Follow Up**
>
> Dear Reviewer xc11,
>
> We hope this message finds you well. We are writing to kindly follow up regarding the rebuttal feedback for our submission. The rebuttal deadline is approaching, and we want to ensure that we address all your comments thoroughly and accurately.
>
> Thank you again for your time and consideration during this busy period.
>
> With kind regards,
> Authors

---

### Official Review · Reviewer_J47W · 2025-11-02

**Soundness:** 3
**Presentation:** 2
**Contribution:** 2
**Rating:** 4
**Confidence:** 3

**Summary:**

•	The paper presents FACTS, an agent-based workflow for query-focused table summarization. The system produces an offline template that combines SQL queries with Jinja2 rendering. The idea is to let LLMs generate reusable templates once and reuse them across tables with the same schema. The workflow has three stages: schema-guided filtering, SQL generation, and text rendering, each verified by an LLM Council. Experiments on FeTaQA, QTSumm, and QFMTS show consistent gains over several prompt-based and agent baselines.

**Strengths:**

•	The separation between template generation and execution is practical, allowing expensive reasoning to occur once and inexpensive SQL execution to repeat many times.
•	The LLM Council improves reliability, as the full FACTS workflow achieves a 100% SQL pass rate, whereas the single-call variant fails on several queries.
•	The method scales efficiently. Once the template is ready, runtime remains nearly constant even as table size increases, while baseline methods become slower.

**Weaknesses:**

•	W1. Section 3.1 suggests that the generated templates can generalize to semantically similar queries, but the evaluation in Section 4.6 only varies table values while keeping both the query and schema fixed.  The results therefore demonstrate reuse under identical conditions rather than true semantic variation.  In addition, all reuse tests are conducted on tables with the same schema, so it is unclear how the method would behave if column names were changed or new fields introduced.

•	W2. The improvements over strong baselines are marginal. In Table 2, FACTS scores 46.0 BLEU and 70.8 ROUGE-L on QFMTS, while SPaGe reaches 45.7 and 68.3. On FeTaQA, SPaGe even performs slightly better.

•	W3. The council design involves several LLMs working together, but the paper only compares the full setup with a single-call variant.  While this shows that iteration helps, it leaves open whether multiple models are actually necessary or if a single model with repeated prompting could achieve similar performance.

•	W4. The evaluation focuses on BLEU, ROUGE-L, METEOR, and SQL pass rate, which assess fluency and executability but not factual consistency.  A summary could read well yet misrepresent the SQL output.  The paper lacks a measure of factual faithfulness or any human verification of whether the rendered summary matches the actual SQL execution results.

•	W5. The paper highlights offline scalability but does not quantify the actual generation cost. For example, Algorithm 2 includes iterative loops in Stage 1 and Stage 2, yet the average number of iterations per template is never reported.

**Questions:**

•	How broadly can a generated template handle query or schema variations? For example, can it adapt to “top 3” vs “top 5,” renamed columns, or added fields without regeneration?
•	Section 3.3 already includes an execute–repair loop. Is there an ablation comparing the three-model council with a single-model version using the same loop?
•	How do the authors ensure that the generated summary is factually consistent with the actual SQL execution? For example, could the Jinja2 template render an incorrect value or variable while still achieving a high BLEU score?
•	What is the average number of iterations (t) in Algorithm 2 for each benchmark? What are the average runtime and token cost for generating a single template?

---

> ### Author Response · Authors · 2025-11-21
>
> We sincerely thank Reviewer J47W for the thoughtful and constructive feedback. We have carefully revised the manuscript according to the reviewer’s comments, and all corresponding changes are highlighted in red in the revised PDF. Below, we provide point-by-point responses to the weaknesses and questions raised by the reviewer.
>
> ## Weakness and Questions
> > W1. Section 3.1 suggests that the generated templates can generalize to semantically similar queries...
>
> > How broadly can a generated template handle query or schema variations? For example, can it adapt to “top 3” vs “top 5,” renamed columns, or added fields without regeneration?
>
> We thank the reviewer for the insightful comment.
> To address the concern about generalization to semantically similar queries, we have added a new experiment in Section 4.6 (lines 484–489) of the revised manuscript.
> In this experiment, we use GPT-5 to paraphrase both the user queries and corresponding reference summaries for 100 randomly sampled QTSumm examples, creating semantically similar but lexically different variants.
> The original offline templates are then applied without regeneration.
> FACTS maintains comparable performance with BLEU = 21.8, ROUGE-L = 43.5, and METEOR = 50.8, confirming that it generalizes robustly to semantic variations of the same query while preserving effectiveness.
> Regarding schema variations, our approach is robust to simple changes such as renamed columns, which can be handled by updating the column names in the generated SQL queries.
> For more substantial schema changes (e.g., added columns), our agentic workflow needs to be executed again to regenerate the offline templates.
> However, as shown in Figure3, the overall latency remains comparable to SPaGe even when the offline template must be generated from scratch.
> Moreover, once the new template is generated, it can be efficiently reused across all tables sharing the same schema and semantically similar queries.
> We consider handling more complex schema drift without regenerating templates beyond the current scope, but note that FACTS remains practical and competitive even when regeneration is required.
>
> > W2. The improvements over strong baselines are marginal...
>
> We thank the reviewer for this comment.
> As stated in the manuscript (lines 412–414), Table 2 shows that FACTS consistently outperforms or matches the strongest baseline SPaGe on all datasets in at least two of the reported metrics, demonstrating solid quantitative performance.
> In the revised manuscript, we further strengthen this conclusion through our newly added human evaluation in Section 4.7 (lines 502–529).
> FACTS achieves 97\% SQL–query intent match, 94\% SQL execution accuracy, 98\% template rendering accuracy, and 3\% Council consensus error rate, confirming its factual reliability.
> Moreover, in the pairwise human preference study, human judges consistently favor FACTS over SPaGe, preferring it in 55\% of cases for completeness, 59\% for correctness, and 60\% for hallucination reduction.
> These results collectively show that FACTS delivers not only competitive automatic scores but also superior factual faithfulness and user-perceived quality compared with SPaGe.
>
> > W3. The council design involves several LLMs working together...
>
> > Section 3.3 already includes an execute–repair loop. Is there an ablation comparing the three-model council with a single-model version using the same loop?
>
> We thank the reviewer for this insightful question.
> To isolate the impact of Council composition, we have added a new experiment in the revised manuscript that evaluates a **FACTS (GPT-Only)** variant, in which all three models in the Council are replaced with GPT-4o-mini (Section 4.3, lines 392–395).
> This setup allows us to assess the effectiveness of FACTS independent of cross-model diversity.
> The results of this variant are reported in Table 2 and analyzed in Section 4.4 (lines 414–420).
> As discussed there, while the full FACTS framework achieves the best overall performance, partly due to the diversity of reasoning behaviors introduced by heterogeneous Council members, the GPT-Only variant remains competitive and still outperforms most baseline methods across all datasets and metrics.
> These findings demonstrate that the core FACTS workflow is intrinsically effective even without model diversity, and that Council heterogeneity further enhances its performance.

---

> ### Author Response · Authors · 2025-11-21
>
> > W4. The evaluation focuses on BLEU, ROUGE-L, METEOR, and SQL pass rate, which assess fluency and executability but not factual consistency...
>
> > How do the authors ensure that the generated summary is factually consistent with the actual SQL execution? For example, could the Jinja2 template render an incorrect value or variable while still achieving a high BLEU score?
>
> We thank the reviewer for this important observation.
> We fully agree that automatic metrics such as BLEU, ROUGE-L, METEOR, and SQL pass rate primarily measure fluency or executability and cannot fully capture factual consistency.
> To address this limitation, we have added a comprehensive human evaluation in Section 4.7 (lines 502–529) of the revised manuscript.
> In this evaluation, we assess 200 randomly sampled examples (100 from QTSumm and 100 from QFMTS) across four aspects:
> (1) whether each generated SQL semantically matches the user query (intent match),
> (2) whether the SQL execution results correctly correspond to the numerical or factual content in the reference summary (SQL execution accuracy),
> (3) whether the numbers and facts rendered in the final summary faithfully reflect the SQL execution results (template rendering accuracy), and
> (4) whether the LLM Council unanimously accepts a specification or SQL query that leads to an incorrect result (Council consensus error rate).
> The Council consensus error rate is computed across two stages: (i) schema-guided specification and filtering, and (ii) SQL query generation.
> While no errors occur during the specification stage, about 6\% of the SQL queries approved by the Council lead to incorrect results during the generation stage, yielding an overall average error rate of approximately 3\%.
> Overall, FACTS achieves 97\% intent match, 94\% SQL execution accuracy, 98\% template rendering accuracy, and 3\% Council consensus error rate, demonstrating strong factual reliability and semantic correctness.
> We note that the SQL execution accuracy is lower than the template rendering accuracy because some SQL queries compute incorrect values with respect to the reference summary, while the rendering accuracy reflects whether the template correctly verbalizes the SQL execution results.
> Therefore, the overall factual correctness of the generated summaries can be estimated as $94\\% \times 98\\% \approx 92\\%$.
> These results directly confirm that the rendered summaries remain faithful to the actual SQL execution outputs and that FACTS ensures factual alignment beyond what automatic metrics can measure.
>
> > W5. The paper highlights offline scalability but does not quantify the actual generation cost...
>
> > What is the average number of iterations (t) in Algorithm 2...
>
> We thank the reviewer for this valuable comment.
> In the revised manuscript, we clarify that the latency reported in Figure 3 already includes the cost of the initial offline template generation.
> We explicitly emphasize this point in Section 4.6 (lines 478–479):
> "We emphasize that the latency reported in Figure 3 already includes the cost of the initial offline template generation."
> Furthermore, we have added a new paragraph on computation cost in Section 4.4 (lines 421–427), which provides detailed quantitative statistics of the iterative process:
> "Across the three datasets evaluated, each sample involves on average 2.47 accepted guiding questions or filtering rules (2.25 initially accepted and 0.22 accepted after one round of revision), 1.36 SQL refinement rounds, and 1.84 template refinement rounds, with the maximum patience set to three rounds in our experiments.
> We further provide token-level efficiency analysis showing that the entire FACTS workflow requires 9,922 input tokens and 1,045 output tokens per sample on average (including all stages and Council outputs), offering a comprehensive view of runtime and generation cost."
> These additions quantify the iterative loops in Algorithm 2 and clearly demonstrate that the overall generation cost is minimal relative to the efficiency gains achieved through offline template reuse.

---

> ### Author Response · Authors · 2025-11-24
> **Looking Forward to Your Feedback**
>
> Dear Reviewer J47W,
>
> We hope this message finds you well. We would like to kindly follow up regarding the rebuttal feedback for our submission. We truly appreciate the time and effort you have devoted to reviewing our paper, and we are happy to provide any clarifications if needed.
>
> We understand that this is a busy period and greatly value your consideration. Please let us know if any further information would be helpful from our side.
>
> With kind regards,
>
> Authors

---

> ### Author Response · Authors · 2025-11-28
> **Follow Up**
>
> Dear Reviewer J47W,
>
> We hope this message finds you well. We are writing to kindly follow up regarding the rebuttal feedback for our submission. The rebuttal deadline is approaching, and we want to ensure that we address all your comments thoroughly and accurately.
>
> Thank you again for your time and consideration during this busy period.
>
> With kind regards,
> Authors

---

### Official Review · Reviewer_c8fF · 2025-11-03

**Soundness:** 3
**Presentation:** 3
**Contribution:** 2
**Rating:** 4
**Confidence:** 4

**Summary:**

This paper presents FACTS, an agentic framework for query-focused table summarization that generates reusable offline templates (SQL + Jinja2) validated by an LLM Council. The approach is evaluated on FeTaQA, QTSumm, and QFMTS, showing strong results compared with recent baselines such as TaPERA (ACL 2024) and SPaGe (2025) under the same model backbone. The method is practical and well-implemented, but its core idea of offline template generation and validation is not fundamentally new, having appeared in other data-to-text and report-generation contexts. Moreover, while the framework is claimed to be efficient, no runtime or cost evaluation is provided to substantiate that claim.

**Strengths:**

The paper is technically solid, clearly written, and experimentally thorough, but its novelty is incremental and key claims about speed and efficiency are not empirically supported.
 It would still be a useful addition for practitioners and researchers interested in reliable, schema-aware table summarization.

- Comprehensive evaluation across all major query-focused summarization benchmarks with up-to-date baselines.
- Methodologically coherent framework combining schema-guided SQL generation, Jinja2 templating, and multi-LLM validation.
- Strong reproducibility — clear prompts, detailed methods, and transparent implementation.
- The design addresses practical aspects such as reusability and privacy.

**Weaknesses:**

- Limited novelty: offline or template-based summarization has prior art; the main contribution is integration rather than conceptual innovation.
- No empirical efficiency evidence: runtime, latency, or token-cost comparisons are missing despite efficiency claims.
- Evaluation metrics (BLEU, ROUGE-L, METEOR) capture surface similarity but not factual consistency or execution accuracy.
- Narrow domain scope: all datasets are Wikipedia-based, so real-world privacy or deployment benefits are not demonstrated.

**Questions:**

- Add quantitative runtime and efficiency measurements.
- Include factual-consistency or SQL execution accuracy metrics.
- Clarify the novelty position as a domain adaptation or integration of existing ideas.
- Consider testing on non-Wikipedia domains to support practical claims.

---

> ### Author Response · Authors · 2025-11-21
>
> We sincerely thank Reviewer c8fF for the thoughtful and constructive feedback. We have carefully revised the manuscript in response to the reviewer’s comments, and all corresponding changes are highlighted in red in the revised PDF. Below, we provide point-by-point responses to the weaknesses and questions raised by the reviewer.
>
> ## Weakness and Questions
> > Limited novelty: offline or template-based summarization has prior art; the main contribution is integration rather than conceptual innovation.
>
> > Clarify the novelty position as a domain adaptation or integration of existing ideas.
>
> We thank the reviewer for this insightful comment.
> We acknowledge that template-based summarization has prior art [1], where templates are manually crafted and then filled with textual content using fixed rules.
> However, our contribution is fundamentally different in both *automation* and *application scope*.
> (1) Manual template construction is highly time-consuming and domain-specific, while FACTS automates this process through an agentic workflow that integrates guided question generation, executable SQL synthesis, and validated Jinja2 template rendering, producing reusable templates automatically.
> (2) Unlike prior text summarization work, our setting involves *query-focused table summarization*, which requires structured reasoning and factual grounding over tabular information.
> To further clarify our contribution, we have added the following statement in the revised manuscript (lines 92–94):
> "To the best of our knowledge, FACTS introduces the first agentic framework that automates offline template generation for query-focused table summarization."
> This addition highlights the conceptual innovation of FACTS beyond trivial integration of existing ideas.
>
> [1] Zhou et al., Template-Filtered Headline Summarization. Text Summarization Branches Out, 2004.
>
> > No empirical efficiency evidence: runtime, latency, or token-cost comparisons are missing despite efficiency claims.
>
> > Add quantitative runtime and efficiency measurements.
>
> We thank the reviewer for this comment.
> In the revised manuscript, we clarify that the latency reported in Figure 3 already includes the cost of the initial offline template generation.
> This clarification has been explicitly added in Section 4.6 (lines 478–479):
> "We emphasize that the latency reported in Figure 3 already includes the cost of the initial offline template generation."
> As shown in the figure, FACTS is slightly slower than Reason-then-Summ on a single table because it must first create the reusable template, but becomes substantially faster once the template is reused across multiple tables with the same schema.
> Furthermore, we provide detailed efficiency analysis in Section 4.4 (lines 421–427), reporting that each sample involves on average 2.47 accepted guiding questions or filtering rules (2.25 initially accepted and 0.22 accepted after one round of revision), 1.36 SQL refinement rounds, and 1.84 template refinement rounds (maximum = 3).
> We also include a token-level analysis showing that the entire FACTS workflow requires approximately 9,922 input tokens and 1,045 output tokens per sample (including all stages and Council outputs), offering a complete view of runtime and computational cost.
> These results demonstrate that the iterative generation cost is minimal compared with the substantial efficiency gains achieved through template reuse.

---

> ### Author Response · Authors · 2025-11-21
>
> > Evaluation metrics (BLEU, ROUGE-L, METEOR) capture surface similarity but not factual consistency or execution accuracy.
>
> > Include factual-consistency or SQL execution accuracy metrics.
>
> We thank the reviewer for this valuable comment.
> We agree that overlap-based metrics such as BLEU, ROUGE-L, and METEOR mainly measure surface similarity and do not fully capture factual consistency or execution accuracy.
> To address this limitation, we have added a comprehensive human evaluation in Section 4.7 (lines 502–529) of the revised manuscript.
> This evaluation explicitly measures four aspects:
> (1) whether each generated SQL semantically matches the user query (intent match),
> (2) whether the SQL execution results correctly correspond to the numerical or factual content in the reference summary (SQL execution accuracy),
> (3) whether the rendered summary faithfully reflects the SQL execution results (template rendering accuracy), and
> (4) whether the LLM Council unanimously accepts a specification or SQL query that leads to an incorrect result (Council consensus error rate).
> The Council consensus error rate is computed across two stages: (i) schema-guided specification and filtering, and (ii) SQL query generation.
> While no errors occur during the specification stage, about 6\% of the SQL queries approved by the Council lead to incorrect results during the generation stage, yielding an overall average error rate of approximately 3\%.
> Overall, FACTS achieves 97\% intent match, 94\% SQL execution accuracy, 98\% template rendering accuracy, and 3\% Council consensus error rate, confirming its factual reliability.
> We note that the SQL execution accuracy is lower than the template rendering accuracy because some SQL queries compute incorrect values with respect to the reference summary, while the rendering accuracy reflects whether the template correctly verbalizes the SQL execution results.
> Therefore, the overall factual correctness of the generated summaries can be estimated as $94\\% \times 98\\% \approx 92\\%$.
> Moreover, we perform a side-by-side human preference study comparing FACTS with the strongest baseline SPaGe, where human judges evaluate completeness, correctness, and hallucination.
> FACTS is preferred in 55\% of cases for completeness, 59\% for correctness, and 60\% for hallucination reduction.
> These findings demonstrate that the factual consistency and correctness of FACTS are validated by human judgment beyond what surface-level automatic metrics can capture.
>
> > Narrow domain scope: all datasets are Wikipedia-based, so real-world privacy or deployment benefits are not demonstrated.
>
> > Consider testing on non-Wikipedia domains to support practical claims.
>
> We thank the reviewer for this comment.
> We follow prior work [1] in adopting FeTaQA, QTSumm, and QFMTS as the standard benchmarks for query-focused table summarization.
> As stated in Section 4.1, while FeTaQA and QTSumm are derived from Wikipedia, QFMTS is not.
> QFMTS is based on the Spider dataset [2], which includes 200 databases spanning 138 distinct domains, such as university courses, online SQL tutorials, textbook examples, and public CSV repositories.
> To clarify this point, we have highlighted this property of QFMTS in the revised manuscript (lines 340–343):
> "QFMTS is based on the Spider dataset, which includes $200$ databases spanning $138$ distinct domains, such as university courses, online SQL tutorials, textbook examples, and public CSV repositories."
> This diversity makes QFMTS a cross-domain dataset, demonstrating that our evaluation is not limited to a narrow domain scope.
> In addition, the privacy advantage of FACTS stems from its design, as no raw cell values are ever sent to external LLMs; thus, its privacy guarantee is orthogonal to the dataset source.
>
> [1] Zhang et al., Beyond Natural Language Plans: Structure-Aware Planning for Query-Focused Table Summarization, 2025.
>
> [2] Yu et al., Spider: A Large-Scale Human-Labeled Dataset for Complex and Cross-Domain Semantic Parsing and Text-to-SQL Task, EMNLP 2018.

---

> ### Author Response · Authors · 2025-11-24
> **Looking Forward to Your Feedback**
>
> Dear Reviewer c8fF,
>
> We hope this message finds you well. We would like to kindly follow up regarding the rebuttal feedback for our submission. We truly appreciate the time and effort you have devoted to reviewing our paper, and we are happy to provide any clarifications if needed.
>
> We understand that this is a busy period and greatly value your consideration. Please let us know if any further information would be helpful from our side.
>
> With kind regards,
>
> Authors

---

> ### Author Response · Authors · 2025-11-28
> **Follow Up**
>
> Dear Reviewer c8fF,
>
> We hope this message finds you well. We are writing to kindly follow up regarding the rebuttal feedback for our submission. The rebuttal deadline is approaching, and we want to ensure that we address all your comments thoroughly and accurately.
>
> Thank you again for your time and consideration during this busy period.
>
> With kind regards,
> Authors

---

### Official Review · Reviewer_CGHL · 2025-11-03

**Soundness:** 2
**Presentation:** 3
**Contribution:** 2
**Rating:** 4
**Confidence:** 4

**Summary:**

The paper proposes the FACTS framework for Query-Focused Table Summarization (QFTS).
The core idea lies in offline template generation: SQL queries are paired with Jinja2 templates to form reusable templates, and an iterative LLM Council process ensures SQL executability and consistency between queries and templates.
The system protects privacy by uploading only table schemas, allowing templates to be reused across tables with the same schema, thus improving efficiency.
Experiments on FeTaQA, QTSumm, and QFMTS show that FACTS achieves superior or near-best BLEU, ROUGE-L, and METEOR scores, and demonstrates strong SQL executability and reasoning stability.

**Strengths:**

- Template reuse significantly reduces repeated inference costs and achieves high efficiency under fixed schemas.

- Generating text based on SQL execution results reduces hallucinations compared with prompt-based methods.

**Weaknesses:**

- Using SQL pass rate only measures syntactic correctness and does not guarantee semantic consistency with user queries; BLEU and similar scores may also be hacked.

- Fixed templates may degrade under schema drift, column name changes, or complex multi-table logic.

- Only latency in the reuse phase is reported; the multi-round iterative cost of initial template generation is not disclosed.

- In multi-table scenarios, potential degradation and execution inconsistencies are not analyzed.

**Questions:**

- How is the semantic consistency between SQL and user queries verified? Is there a plan to introduce execution-result-to-summary factual alignment metrics?

- When the schema changes slightly, how can template reuse be maintained?

- Can the authors report the full generation cost, including iteration rounds and token usage?

- The Council’s “majority voting + self-correction” logic, under a self-bootstrapping setup without ground truth, may converge to consistent errors — does this occur, and how often?

If authors can address my concern especially in evaluation, I'm free to raise my score.

---

> ### Author Response · Authors · 2025-11-21
>
> We sincerely thank Reviewer CGHL for the thoughtful and constructive feedback. We have carefully revised the manuscript according to the reviewer’s comments, and all corresponding changes are highlighted in red in the revised PDF. Below we provide point-by-point responses to the weaknesses and questions raised by the reviewer.
>
> ## Weakness and Questions
> > Using SQL pass rate only measures syntactic correctness and does not guarantee semantic consistency with user queries; BLEU and similar scores may also be hacked.
>
> > How is the semantic consistency between SQL and user queries verified? Is there a plan to introduce execution-result-to-summary factual alignment metrics?
>
> We thank the reviewer for pointing out that SQL pass rate alone mainly reflects syntactic validity rather than semantic correctness.
> In the revised manuscript, we add a comprehensive human evaluation in Section4.7 (lines 502–529) to explicitly assess four aspects:
> (1) whether each generated SQL semantically matches the user query (intent match),
> (2) whether the SQL execution results correctly correspond to the numerical or factual content in the reference summary (SQL execution accuracy),
> (3) whether the numbers and facts rendered in the final summary faithfully reflect the SQL execution results (template rendering accuracy), and
> (4) whether the LLM Council unanimously accepts a specification or SQL query that later produces an incorrect result (Council consensus error rate).
> The Council consensus error rate is computed across two stages: (i) schema-guided specification and filtering, and (ii) SQL query generation.
> While no errors occur during the specification stage, about 6\% of the SQL queries approved by the Council lead to incorrect results during the generation stage, yielding an overall average error rate of approximately 3\%.
> Overall, FACTS achieves 97\% intent match, 94\% SQL execution accuracy, 98\% template rendering accuracy, and a very low Council consensus error rate of about 3\%, demonstrating strong semantic consistency, factual reliability, and robustness beyond syntactic correctness.
> We note that the SQL execution accuracy is lower than the template rendering accuracy because some SQL queries compute incorrect values with respect to the reference summary, while the rendering accuracy reflects whether the template correctly verbalizes the SQL execution results.
> Therefore, the overall factual correctness of the generated summaries can be estimated as $94\\% \times 98\\% \approx 92\\%$.
> Although FACTS achieves high factual accuracy, the automatic metrics primarily capture surface-level overlap and semantic similarity rather than factual correctness.
> Our human evaluation thus provides complementary insights beyond what automatic metrics can measure.
>
> > Fixed templates may degrade under schema drift, column name changes, or complex multi-table logic.
>
> > When the schema changes slightly, how can template reuse be maintained?
>
> We appreciate the reviewer’s comment.
> As stated in the original manuscript (Section 3.1, lines 207–210), our method is explicitly designed for reusable offline templates under the same schema: ``Once generated, the same offline template can be directly applied to any table sharing the same schema and answering the same user query or semantically similar queries.''
> To further emphasize this design choice, in the revised manuscript we have added the following clarification in lines 210–211:
> "In this work, we define template reusability under an identical schema, without considering schema drift or renamed columns."
> Regarding schema variations, our approach is robust to simple changes such as renamed columns, which can be handled by updating the column names in the generated SQL queries.
> For more substantial schema changes (e.g., added columns), our agentic workflow needs to be executed again to regenerate the offline templates.
> However, as shown in Figure3, the overall latency remains comparable to SPaGe even when the offline template must be generated from scratch.
> Moreover, once the new template is generated, it can be efficiently reused across all tables sharing the same schema and semantically similar queries.
> We consider handling more complex schema drift without regenerating templates beyond the current scope, but note that FACTS remains practical and competitive even when regeneration is required.
> Regarding complex multi-table reasoning, our QFMTS benchmark already includes queries spanning multiple tables, where FACTS achieves performance that is competitive with or superior to other baselines, demonstrating its effectiveness in handling multi-table scenarios.

---

> ### Author Response · Authors · 2025-11-21
>
> > Only latency in the reuse phase is reported; the multi-round iterative cost of initial template generation is not disclosed.
>
> > Can the authors report the full generation cost, including iteration rounds and token usage?
>
> We thank the reviewer for raising this point.
> As clarified in the revised manuscript, the latency reported in Figure 3 of the original version already includes the cost of the initial offline template generation.
> In the revised version, we explicitly emphasize this in Section 4.6 (lines 478–479):
> "We emphasize that the latency reported in Figure 3 already includes the cost of the initial offline template generation."
> As shown in the figure, with a single table, FACTS is slightly slower than Reason-then-Summ because it must first create the reusable template, and is comparable to SPaGe.
> However, once the template is generated, FACTS achieves a substantial speed advantage when reusing it across multiple tables with the same schema.
> In addition, we have added a new paragraph on computation cost in Section 4.4 that provides detailed statistics of the iterative process (lines 421–427): ``Across the three datasets evaluated, each sample involves on average $2.47$ accepted guiding questions or filtering rules ($2.25$ initially accepted and $0.22$ accepted after one round of revision), $1.36$ SQL refinement rounds, and $1.84$ template refinement rounds, with the maximum patience set to three rounds in our experiments.
> We further provide token-level efficiency analysis showing that the entire FACTS workflow requires $9,922$ input tokens and $1,045$ output tokens per sample on average (including all stages and Council outputs), offering a comprehensive view of runtime and generation cost."
> These additions clearly demonstrate that the iterative generation cost is minimal relative to the efficiency gains from reuse.
>
> > In multi-table scenarios, potential degradation and execution inconsistencies are not analyzed.
>
> We thank the reviewer for the comment.
> The QFMTS dataset already involves multi-table scenarios, and FACTS consistently outperforms or is competitive with baselines (Table 2).
> To further verify robustness under such settings, we have added a new human evaluation in Section 4.7 (lines 502–529) of the revised manuscript, which jointly assesses performance on both single-table (QTSumm) and multi-table (QFMTS) datasets.
> In this evaluation, we measure SQL–query semantic consistency, SQL execution accuracy, template rendering accuracy, and Council consensus error rate on 200 randomly sampled examples (100 from each dataset).
> FACTS achieves 97\% intent match, 94\% SQL execution accuracy, 98\% template rendering accuracy, and 3\% Council consensus error rate, indicating high factual reliability without noticeable degradation when reasoning over multiple tables.
> We note that the SQL execution accuracy is lower than the template rendering accuracy because some SQL queries compute incorrect values with respect to the reference summary, while the rendering accuracy reflects whether the template correctly verbalizes the SQL execution results.
> Therefore, the overall factual correctness of the generated summaries can be estimated as $94\% \times 98\% \approx 92\%$.
> Moreover, we conduct a side-by-side human preference study comparing FACTS with the strongest baseline SPaGe on the same two datasets, where human judges evaluate completeness, correctness, and hallucination.
> FACTS is preferred in 55\% of cases for completeness, 59\% for correctness, and 60\% for hallucination reduction, demonstrating its superior factual consistency and reliability across both single- and multi-table summarization tasks.
>
> > The Council’s “majority voting + self-correction” logic, under a self-bootstrapping setup without ground truth, may converge to consistent errors — does this occur, and how often?
>
> We thank the reviewer for raising this insightful question.
> This issue is examined directly in our new human evaluation presented in Section 4.7 (lines 502–529) of the revised manuscript.
> During the human evaluation, we specifically check cases where the LLM Council unanimously accepts a specification or SQL query that leads to an incorrect result later (Council consensus error rate).
> The Council consensus error rate is computed across two stages: (i) schema-guided specification and filtering, and (ii) SQL query generation.
> While no errors occur during the specification stage, about 6\% of the SQL queries approved by the Council lead to incorrect results during the generation stage, yielding an overall average error rate of approximately 3\%, indicating that the Council’s majority-voting process seldom amplifies systematic errors or converges to incorrect outputs.

---

> ### Author Response · Authors · 2025-11-24
> **Looking Forward to Your Feedback**
>
> Dear Reviewer CGHL,
>
> We hope this message finds you well. We would like to kindly follow up regarding the rebuttal feedback for our submission. We truly appreciate the time and effort you have devoted to reviewing our paper, and we are happy to provide any clarifications if needed.
>
> We understand that this is a busy period and greatly value your consideration. Please let us know if any further information would be helpful from our side.
>
> With kind regards,
>
> Authors

---

> ### Author Response · Authors · 2025-11-28
> **Follow Up**
>
> Dear Reviewer CGHL,
>
> We hope this message finds you well. We are writing to kindly follow up regarding the rebuttal feedback for our submission. The rebuttal deadline is approaching, and we want to ensure that we address all your comments thoroughly and accurately.
>
> Thank you again for your time and consideration during this busy period.
>
> With kind regards,
> Authors

---

### Author Response · Authors · 2025-12-03
**Rebuttal Summary**

We thank the AC for handling our submission and for coordinating the detailed reviews. Below we briefly summarize the consensus strengths and how our revision addresses the key concerns.

**Reviewers generally agree that FACTS is a practical and well-implemented framework for query-focused table summarization: it separates SQL-based computation from Jinja2 surface realization, enables reusable offline templates tied to table schemas, achieves strong or near-best scores on FeTaQA, QTSumm, and QFMTS, and offers clear advantages in reusability, scalability, and privacy (only schemas, not cell values, are sent to LLMs).** They also highlight the methodological coherence of the three-stage agentic pipeline with an LLM Council, the 100% SQL pass rate vs. single-call variants, and the clarity and reproducibility of the paper (prompts, pseudocode, case studies).

Most concerns focus on evaluation (factuality, efficiency, council behavior), claims about generalization and privacy, and the precise scope of template reuse and novelty. In the revision, we:
- Add a comprehensive **human evaluation** (Sec. 4.7) over 200 examples, measuring SQL–query intent match, SQL execution accuracy, template–result alignment, and Council consensus error rate. FACTS attains 97% intent match, 94% SQL execution accuracy, 98% template rendering accuracy, and ≈3% Council consensus error, implying ≈92% overall factual correctness and directly addressing worries that summaries might “read well yet misrepresent” SQL outputs.
- Report a **side-by-side human preference study** (FACTS vs. SPaGe) on QTSumm and QFMTS, where annotators judge completeness, correctness, and hallucination. FACTS is preferred in 55% (completeness), 59% (correctness), and 60% (hallucination), complementing the small but consistent automatic-metric gains and showing that our approach is perceived as more factually reliable.
- Provide **detailed efficiency and cost analysis**: we clarify that Figure 3’s latency numbers already include initial template creation and all Council iterations, and we now report average rounds (2.47 accepted guiding questions/rules, 1.36 SQL refinements, 1.84 template refinements per sample, max 3 each) and token usage (≈9,922 input and 1,045 output tokens per sample including all stages and Council calls). This gives a full accounting of the amortized cost and supports the efficiency claims.
- Clarify the **scope and generalization of template reuse**: (i) we make explicit that our main reusability setting assumes an identical schema; renamed columns can be handled by updating SQL, while substantial schema drift requires regenerating templates (with still-competitive latency), and (ii) we add a new experiment where we paraphrase 100 QTSumm queries and reference summaries and reuse the original templates without regeneration, observing comparable BLEU/ROUGE/METEOR, thus demonstrating robustness to semantic query variations.
- Strengthen the **Council and ablation story**: beyond the original Single-Call variant, we add a **FACTS (GPT-only)** variant where all Council members are the same model. This variant remains competitive and outperforms most baselines, while the heterogeneous Council achieves the best overall results. Together with the new Council consensus error statistics, this shows that iterative validation is crucial and that model diversity further helps but is not strictly required.
- Address **novelty and domain/practicality concerns**: we clarify that FACTS is, to our knowledge, the first agentic framework that automates offline template generation specifically for query-focused table summarization, combining schema-guided SQL and reusable Jinja2 templates under an LLM-validated workflow. We also emphasize that QFMTS is built on Spider and spans 200 databases across 138 domains, so our evaluation goes beyond Wikipedia-only settings.
- Make the **privacy and deployment assumptions explicit** by adding a formalized threat model in the appendix: FACTS treats the LLM as an external API that only sees schemas and queries, while all data values and SQL execution happen locally. We discuss what is and is not protected under this model and point to possible future extensions (e.g., schema abstraction, obfuscation).

**We hope the AC will consider these revisions, additional experiments, and clarifications when making the final decision. We sincerely appreciate the AC’s time, effort, and service to the community.**

---

### Note · Authors · 2026-01-05

I have read and agree with the venue's withdrawal policy on behalf of myself and my co-authors.